# Leptin-activated hypothalamic BNC2 neurons acutely suppress food intake

Han L. Tan[1], Luping Yin[2], Yuqi Tan[3], Jessica Ivanov[1], Kaja Plucinska[4], Anoj Ilanges[1], Brian R. Herb[5], Putianqi Wang[1], Christin Kosse[1], Paul Cohen[4], Dayu Lin[2] & Jeffrey M. Friedman[1]✉

Leptin is an adipose tissue hormone that maintains homeostatic control of adipose tissue mass by regulating the activity of specific neural populations controlling appetite and metabolism[1]. Leptin regulates food intake by inhibiting orexigenic agouti-related protein (AGRP) neurons and activating anorexigenic pro-opiomelanocortin (POMC) neurons[2]. However, whereas AGRP neurons regulate food intake on a rapid time scale, acute activation of POMC neurons has only a minimal effect[3–5]. This has raised the possibility that there is a heretofore unidentified leptin-regulated neural population that rapidly suppresses appetite. Here we report the discovery of a new population of leptin-target neurons expressing basonuclin 2 (*Bnc2*) in the arcuate nucleus that acutely suppress appetite by directly inhibiting AGRP neurons. Opposite to the effect of AGRP activation, BNC2 neuronal activation elicited a place preference indicative of positive valence in hungry but not fed mice. The activity of BNC2 neurons is modulated by leptin, sensory food cues and nutritional status. Finally, deleting leptin receptors in BNC2 neurons caused marked hyperphagia and obesity, similar to that observed in a leptin receptor knockout in AGRP neurons. These data indicate that BNC2-expressing neurons are a key component of the neural circuit that maintains energy balance, thus filling an important gap in our understanding of the regulation of food intake and leptin action.

Leptin is an adipose tissue-derived hormone that functions as the afferent signal in a negative feedback loop that maintains homeostatic control of adipose tissue mass[1,6]. Leptin regulates food intake and maintains energy balance, in part by inhibiting orexigenic agouti-related protein (AGRP)/neuropeptide Y (NPY) neurons, expressing leptin receptor (LepR), and activating anorexigenic pro-opiomelanocortin (POMC) neurons, which also express LepR[1,2]. These neurons are located in the arcuate nucleus (ARC) of the hypothalamus. POMC is a protein precursor that is cleaved proteolytically to generate α-melanocyte stimulating hormone (αMSH), which reduces food intake by activating the melanocortin 4 receptor (MC4R)[7]. AGRP/NPY neurons increase food intake through projections to many of the same sites as POMC neurons where the AGRP protein diminishes αMSH signalling at the MC4R[7]. AGRP/NPY neurons also directly inhibit POMC neurons[8]. These findings have indicated that food intake is reciprocally regulated by these two neural populations. This proposed 'yin–yang' dynamic between these neural populations is a consistent feature in nearly all models of how the homeostatic control of food intake and body weight is maintained[2]. However, the functional effects and dynamics of POMC and AGRP/NPY neurons diverge in several important respects and several lines of evidence have indicated that there may be other leptin-responsive populations that are crucial for leptin-driven control of food intake and body weight[9,10].

First, while activating AGRP/NPY neurons rapidly leads to food seeking and consumption, activating POMC neurons has only a minimal effect on acute food intake[3–5]. Second, mutations in POMC, POMC processing enzymes or the MC4R cause obesity[7,11], whereas mutations in AGRP and NPY have not been reported to alter body weight, although ablation of these neurons in adulthood has been reported to cause anorexia[12–16]. In addition, while activation of AGRP/NPY neurons is associated with negative valence, POMC neuron activation is not associated with positive valence and also elicits negative valence[17,18]. The role of these populations in leptin signalling has also been evaluated and, whereas a knockout of the LepR in AGRP/NPY neurons of adult mice causes extreme obesity, deletion of the LepR from adult POMC neurons has only a minimal effect[19]. Finally, acute activation of AGRP/NPY neurons alters glucose metabolism—a response not observed with acute manipulation of POMC neurons[20]. Thus, while POMC and AGRP/NPY neurons have opposite effects on weight, their effects are not equivalent in other respects, indicating that they are not precise counterparts. This has raised the possibility that there might be a missing population of LepR-expressing neurons that acutely suppress food intake with a timecourse similar to the effects of AGRP/NPY neurons. Indeed, a previous report indicated that there is a missing population of GABAergic neurons that express LepR and regulate weight[9].

To address this, we conducted a systematic profiling of the transcriptomes of ARC neurons using single-nucleus RNA sequencing (snRNA-seq) and identified a new population of LepR-expressing ARC neurons that co-express the basonuclin 2 (*Bnc2*) gene. Further studies

[1]Laboratory of Molecular Genetics, Howard Hughes Medical Institute, The Rockefeller University, New York, NY, USA. [2]Department of Psychiatry, Neuroscience Institute, New York University Langone Medical Center, New York, NY, USA. [3]Department of Microbiology and Immunology, Stanford University School of Medicine, Stanford, CA, USA. [4]Laboratory of Molecular Metabolism, The Rockefeller University, New York, NY, USA. [5]Department of Pharmacology, Institute for Genome Sciences, University of Maryland School of Medicine, Baltimore, MD, USA. ✉e-mail: friedj@rockefeller.edu

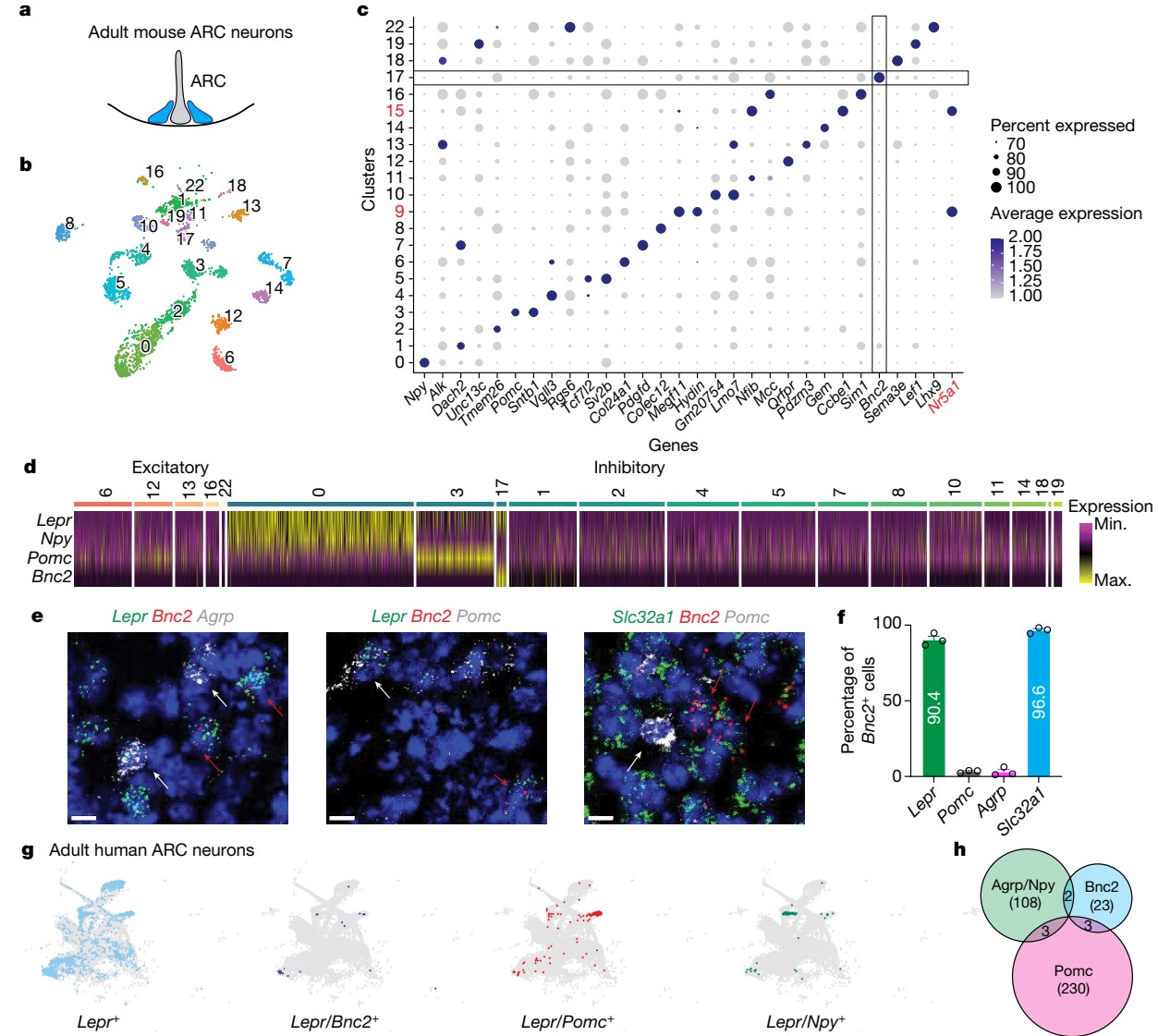

**Fig. 1 | Identification of new LepR-expressing neurons in the ARC.**
**a**, Micro-dissection of mouse ARC. **b**, Uniform manifold approximation and projection (UMAP) plot of neuronal nuclei from the male ARC (*n* = 6 adult wild-type (WT) mice). **c**, Dotplot showing marker genes of individual neuronal clusters, including the *Nr5a1* gene. **d**, Heatmap illustrating the expression of indicated genes across clusters. Neuronal clusters are categorized as excitatory (*Slc17a6* positive) and inhibitory (*Slc321a/Gad1/Gad2* positive). **e**, RNA ISH of *Lepr, Bnc2, Agrp, Pomc* and *Slc32a1* in the ARC of adult male WT mice (*n* = 3 mice). White arrows, *Agrp*- or *Pomc*-positive cells; red arrows, *Bnc2*-positive cells. **f**, Percentage of *Bnc2*-positive cells co-expressing *Lepr, Pomc, Agrp* and *Slc32a1* (*n* = 3 mice). **g**, UMAP plots of *Lepr/Pomc, Lepr/Bnc2* and *Lepr/Npy* neurons in adult human ARC neurons. **h**, Co-localization of *Lepr/Bnc2*, *Lepr/Agrp/Npy* and *Lepr/Pomc* neurons in adult human ARC neurons. Data are presented as mean ± s.e.m. Scale bar, 10 μm. Max., maximum; min., minimum.

showed that BNC2 neurons rapidly induce satiety with similar kinetics to the activation of feeding after AGRP/NPY neural activation. This discovery adds an important new cellular component to the neural network that regulates hunger and satiety and sheds new light on how leptin regulates energy balance.

## Identification of BNC2/LepR neurons

We performed snRNA-seq on ARC that was micro-dissected from adult male mice (Fig. 1a). Nuclei were isolated, stained with a fluorescent antibody to NeuN—a specific marker of neuronal nuclei—and subjected to fluorescence-activated sorting followed by single nucleus sequencing using the 10x single-cell analysis platform[21]. A total of 3,557 cells were profiled, of which 3,481 were identified as neurons on the basis of expression levels of neuronal markers including *Tubb3*, *Snap25*, *Syt1* and *Elavl4*, and the absence of non-neuronal markers (Extended Data Fig. 1a,b). Leiden clustering analysis identified 21 distinct neuron

clusters (Fig. 1b,c). To confirm that these neurons were from the ARC rather than from adjacent regions, we analysed the clusters for specific marker genes that define known ARC populations. This validation process confirmed that 19 of the clusters were indeed located within the ARC with the exception of Clusters 9 and 15, which express *Nr5a1*, indicating their origin from the region of the adjacent ventromedial hypothalamus[22,23] (Fig. 1c). These clusters included one that expresses *Agrp/Npy* (Cluster 0), two expressing *Pomc* (Clusters 3 and 12) neurons and 16 more clusters (Fig. 1d). Clusters 0 and 3 expressed the *Lepr*, as did Cluster 17, which did not express *Agrp*, *Npy* or *Pomc* but instead showed specific expression of the *Bnc2* gene (Fig. 1d). Several of the genes in BNC2 Cluster 17 including *Lepr* were also found in a single cluster (n11.*Trh/Cxcl12*) in a previously published dataset of ARC neuron types[22] (Extended Data Fig. 1c). However, *Bnc2* was not reported as being expressed in this cluster and thyrotropin-releasing hormone (*Trh*) gene—encoding the canonical marker TRH—was also expressed in other clusters that did not express *Lepr*[22]. Thus, BNC2 serves as

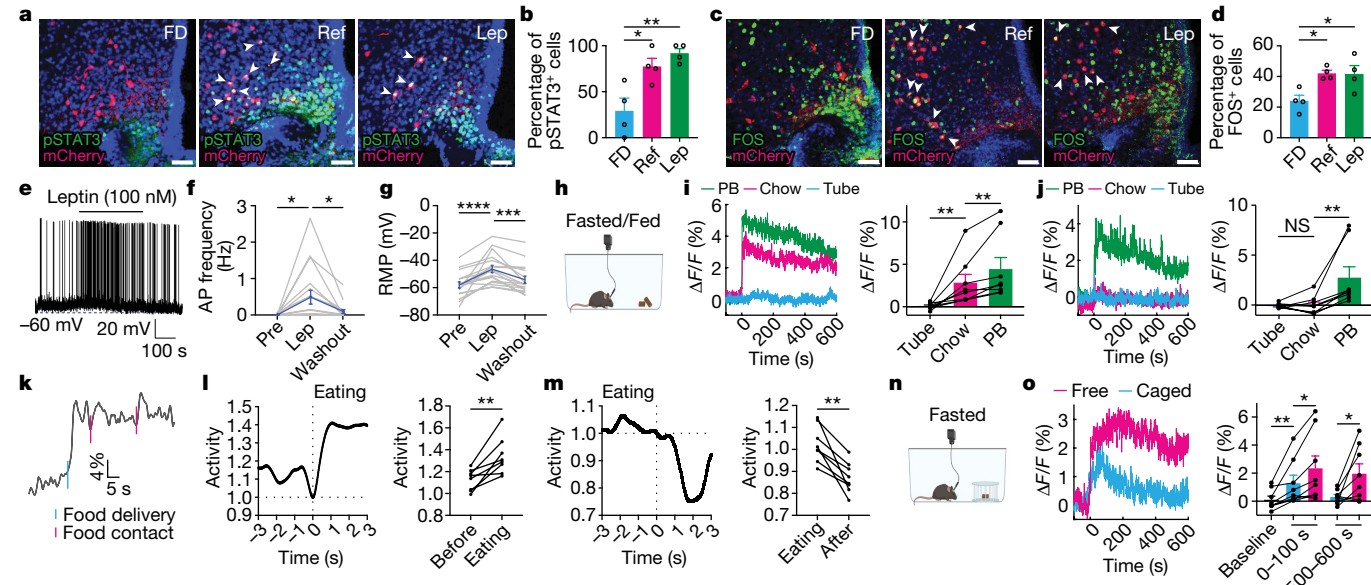

**Fig. 2 | Activation of BNC2 neurons by feeding. a,c**, Representative pSTAT3 (**a**) and FOS (**c**) immunostaining in male mice under three conditions: overnight fasting (FD), fasting followed by 3-h refeeding with chow (Ref) and fasting followed by leptin injection with 3-h wait (Lep) (*n* = 4 mice per group). Arrowheads indicate double-positive cells. **b,d**, Percentage of pSTAT3/mCherry (**b**) and FOS/mCherry (**d**) double-positive cells in mCherry-positive cells (*n* = 4 mice per group, one-way analysis of variance (ANOVA)). **e**, Spontaneous APs of ARC BNC2 neurons from overnight-fasted adult male BNC2-Cre mice injected with AAV-DIO-GFP. Leptin was added at the time indicated. **f,g**, Spontaneous AP frequency (**f**) and resting membrane potential (RMP, **g**) before, during and after leptin application (*n* = 17 cells from four mice per group, Friedman test). **h**–**k**, Calcium recordings from male mice expressing GCaMP6s in BNC2 neurons exposed to an inedible plastic tube, chow or PB (**h**). **i**, Average calcium traces in fasted mice aligned to presentation (left) and fluorescence quantification

(0–300 s) (right, *n* = 8 mice per group, paired Wilcoxon test). **j**, Average calcium traces in mice fed ad libitum aligned to presentation (left) and quantification (0–300 s) (right, *n* = 8 mice per group, paired Student's *t*-test or Wilcoxon test). **k**, Individual calcium trace from an overnight-fasted mouse presented with PB. **l,m**, Average calcium traces in fasted mice presented with PB, before (**l**) and after (**m**) eating, aligned to eating (left) and quantification (right, *n* = 8 mice per group, paired Student's *t*-test). **n,o**, Recordings from fasted male mice presented with chow inside or outside a container (**n**). **o**, Average calcium traces aligned to presentation (left) and quantification (0–100 and 500–600 s) (right, *n* = 8 mice per group, paired Student's *t*-test or Wilcoxon test). Data are presented as mean ± s.e.m. NS, not significant; *$P < 0.05$; **$P < 0.01$; ***$P < 0.001$; ****$P < 0.0001$. Statistical details in Source Data. Scale bars, 50 μm. Illustrations in **h** and **n** were created using BioRender (https://biorender.com).

a specific marker for a new *Lepr* cluster that does not include *Agrp*, *Npy* or *Pomc*.

The total number of cells in the BNC2 cluster (Cluster 17, *n* = 35) was lower than those in the AGRP cluster (Cluster 0, *n* = 650) or the POMC cluster (Cluster 3, *n* = 271). However, the level of LepR expression in this cluster was significantly higher than in the POMC cluster and equivalent to that in the AGRP cluster (Cluster 0, 1.32; Cluster 3, 0.52; Cluster 17, 1.09). We confirmed the co-expression of *Lepr* and *Bnc2* using multiplex in situ hybridization (ISH) of adult mouse hypothalamus. The results showed that more than 90% of *Bnc2* cells in the ARC co-expressed *Lepr* (90.4%) and *Slc32a1* (96.6%)—a marker for inhibitory GABAergic neurons—whereas less than 5% co-expressed either *Agrp* (2.6%) and *Pomc* (3.3%) (Fig. 1e,f). We also determined whether *Bnc2* co-localized with *Lepr* in human hypothalamus ARC[24] and found a significant overlap between *Bnc2* and *Lepr* (Fisher's exact test; *P* = 0.0218; Fig. 1g). Here again, the LepR/BNC2 cells in human hypothalamus were distinct from AGRP/NPY and POMC neurons, as 82.1% (23 of 28) of *Lepr*/*Bnc2* neurons were not co-localized with *Agrp*, *Npy* or *Pomc* (Fig. 1h). Altogether, these results show that BNC2 neurons are a new population of LepR-expressing neurons in mouse and human that does not overlap with the previously characterized LepR AGRP/NPY and POMC populations.

## Activation of BNC2 neurons by feeding

To further study the dynamics and function of BNC2 neurons in ARC, we generated a BNC2-P2A-iCre knockin mouse line (referred to hereafter as BNC2-Cre) by inserting a P2a-iCre fusion in frame at the C terminus of the BNC2 protein (Extended Data Fig. 2a,b). The correct insertion site of iCre was confirmed by PCR of genomic DNA (Extended Data Fig. 2c).

Eutopic expression of Cre in BNC2 neurons in ARC was verified by injecting an adeno-associated virus (AAV) with a Cre-dependent mCherry into the ARC of adult BNC2-Cre mice. ISH showed nearly complete co-localization of *mCherry* and *Bnc2* RNA (Extended Data Fig. 2d). Leptin binding to LepR activates Janus tyrosine kinase 2 (JAK2), leading to STAT3 phosphorylation—a canonical marker of leptin activation[25]—and we confirmed leptin signalling in BNC2 neurons by immunostaining for pSTAT3 after leptin treatment. An AAV-DIO-mCherry was injected into the ARC of adult BNC2-Cre mice and animals subsequently received a single injection of leptin (3 mg kg$^{-1}$) after an overnight fast. This resulted in a significant increase in pSTAT3 in mCherry-labelled BNC2 neurons at 3 h, which was not seen in control animals that received PBS injections (Fig. 2a,b). Refeeding mice after an overnight fast also led to a significant increase of pSTAT3 amounts in BNC2 neurons (Fig. 2a,b). Consistent with this, we also found that the amounts of FOS—a neuron activity marker[26]—increased significantly in BNC2 neurons following refeeding or leptin injection (Fig. 2c,d).

We then performed whole-cell patch-clamp recordings in hypothalamic slices prepared from adult male BNC2-Cre mice injected with an AAV-DIO-EGFP in the ARC. Application of leptin (100 nM) to brain slices prepared from mice that were either fasted or fed ad libitum resulted in a significant depolarization and a substantial increase in action potential (AP) firing rates in green fluorescent protein (GFP)-labelled BNC2 neurons (Fig. 2e–g and Extended Data Fig. 3a–d). These effects were reversible after leptin washout (Fig. 2e–g and Extended Data Fig. 3a–d). In addition, a higher percentage of cells responded to leptin in the slices prepared after an overnight fast (16 of 17) compared with those responding in slices prepared from mice fed ad libitum (7 of 12) (Extended Data Fig. 3e). The greater effect of leptin on neurons from

fasted hypothalamus is consistent with the FOS data showing that the baseline activity of BNC2 neurons is lower after an overnight fast (Fig. 2d).

To assess the in vivo responses, we used fibre photometry to record the effect of food cues on the activity of BNC2 neurons. We injected AAV-DIO-GCaMP6s into the ARC of adult male BNC2-Cre mice and implanted a fibre above the ARC to record neural activity. Mice were fasted overnight and then given access to one of the following: an inedible plastic tube, a pellet of standard chow or peanut butter (PB), all of which they had been exposed to previously. The inedible tube did not have a discernible effect on BNC2 neuron activity (Fig. 2h,i and Extended Data Fig. 4a–c). However, even before consumption, relative to the plastic tube, exposure to standard chow robustly activated BNC2 neurons in seconds (Fig. 2h,i and Extended Data Fig. 4a–c) and exposure to PB, which has a significantly higher fat content, elicited a significantly greater response than did chow (Fig. 2h,i and Extended Data Fig. 4a–c). By contrast, BNC2 neurons from mice fed ad libitum showed little or no response to the inedible tube and standard chow (Fig. 2j and Extended Data Fig. 4d–f) while PB still significantly activated BNC2 neurons (Fig. 2j and Extended Data Fig. 4d–f). These results show that, in fasted mice, the activity of BNC2 neurons is increased by the presence of food and that this is further augmented by palatability and is sensitive to nutritional state (that is, fed versus fasted).

We further noted that the increased fluorescence of BNC2 neurons preceded the initiation of food intake, indicating that they were activated by sensory cues (Fig. 2k and Supplementary Video 1). To assess the impact of food consumption, we analysed the changes in BNC2 neuron activity during the consumption phase. Our analysis showed that BNC2 neuron activity further increased during food consumption compared with its level when animals were exposed to food cues but had not yet consumed it (Fig. 2l,m). These data indicate that BNC2 neurons respond to food cues as well as signals generated during food consumption. To directly test the effect of sensory cues on BNC2 neuron activity, we presented fasted mice with standard chow in a container that allowed them to see and smell the pellet without being able to consume it (Fig. 2n). Under these conditions, BNC2 neurons still showed significant activation, albeit of lesser magnitude and duration compared with the experiments in which the food was not caged (Fig. 2o and Extended Data Fig. 4g,h). This indicates that BNC2 neurons respond to sensory cues and that accessibility to food and its consumption further increases the strength and duration of neural activation. To test whether the sensory-dependent regulation of BNC2 neuron activity depends on experience, we presented new foods, such as PB and sucrose tablets, to naive mice that had never encountered them before after an overnight fast. In this case, we did not observe significant changes in BNC2 neuron activity in the naive mice after the presentation of the food (Extended Data Fig. 4i–l). However, after a delay, the animals began to consume the food, at which point the activity of BNC2 neurons significantly increased (Extended Data Fig. 4i–l). Collectively, these results indicate that BNC2 neurons respond to food-related sensory cues in an experience-dependent manner and that food consumption further activates these neurons.

Finally, we examined whether the removal of food extinguished the neural activation seen after food presentation. In this experiment, mice that had been fasted overnight were presented with accessible chow which was removed 2 or 10 min after it was provided. Removal of food led to a rapid decrease in the activity of BNC2 neurons (Extended Data Fig. 4m–p). By contrast, the BNC2 neurons remained active when food was continuously present (Extended Data Fig. 4m–p).

## BNC2 neurons acutely suppress appetite

We next assessed the function of BNC2 neurons using chemogenetics and optogenetics. AAVs expressing a Cre-dependent stimulatory hM3Dq (Gq-coupled human muscarinic M3 designer receptors exclusively activated by designer drugs, DREADDs) or an inhibitory hM4Di

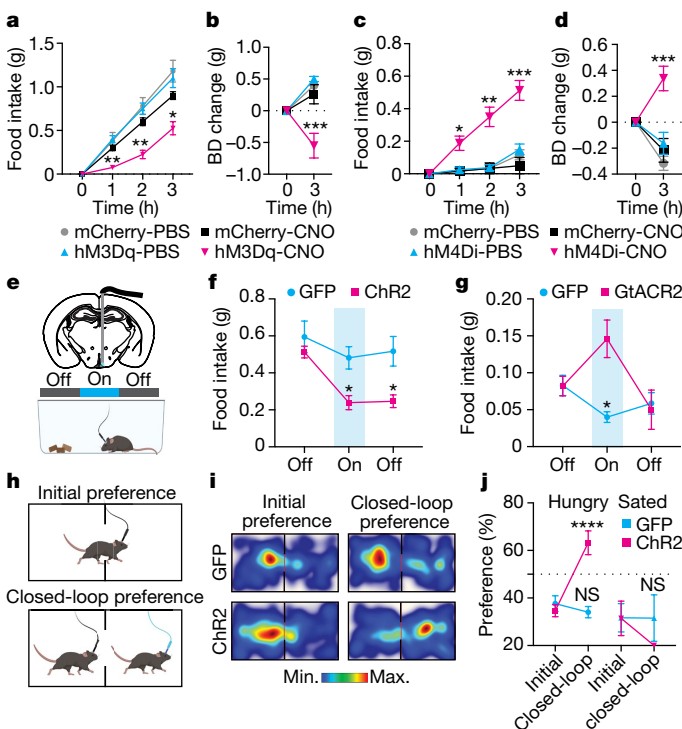

**Fig. 3 | Rapid and sustained satiety driven by BNC2 neurons. a,b,** Hourly food intake (**a**) and body weight (BD) change at 3 h (**b**) in male mice receiving PBS or CNO injection at the onset of the night dark phase ($n = 5$ mCherry-injected mice, $n = 4$ hM3Dq-injected mice; two-way ANOVA). **c,d,** Hourly food intake (**c**) and body weight change at 3 h (**d**) in male mice receiving PBS or CNO injection during the daytime ($n = 6$ mCherry-injected mice, $n = 8$ hM4Di-injected mice; two-way ANOVA). **e,** Unilateral AAV virus injection into the ARC region of adult male BNC2-Cre mice followed by fibre optic implant. **f,** Food intake before, during and after laser stimulation in overnight-fasted mice ($n = 7$ GFP-injected mice, $n = 12$ ChR2-injected mice; two-way ANOVA). **g,** Food intake before, during and after laser stimulation in mice fed ad libitum during the daytime ($n = 7$ GFP-injected mice, $n = 5$ GtACR2-injected mice; two-way ANOVA). **h–j,** Closed-loop place preference (**h,i**) assay to evaluate the valence of BNC2 neurons. **i,** Representative heatmaps of time spent in each chamber by overnight-fasted adult male BNC2-Cre mice injected with GFP or ChR2 during the initial phase and the closed-loop paired phase. **j,** Quantification of place preference (time spent in the pair-stimulated chamber, $n = 6$ GFP-injected mice, $n = 8$ ChR2-injected mice; two-way ANOVA). Data are presented as mean ± s.e.m. NS, not significant; *$P < 0.05$; **$P < 0.01$; ***$P < 0.001$; ****$P < 0.0001$. Statistical details in Source Data. Illustrations in **e** and **h** were created using BioRender (https://biorender.com).

DREADD were injected into the ARC of BNC2-Cre mice (Extended Data Fig. 5a). Mice receiving the AAV-DIO-mCherry were used as controls. We then selectively activated or inhibited these neurons using clozapine-*N*-oxide (CNO)[27]. Activating BNC2 neurons at the beginning of the dark cycle resulted in a significant reduction in food intake and body weight compared with control mice of both sexes (Fig. 3a,b and Extended Data Fig. 5c,d). By contrast, silencing BNC2 neurons during the light cycle significantly increased food consumption and body weight in both male and female mice (Fig. 3c,d and Extended Data Fig. 5e,f).

We further evaluated the dynamics of the feeding response using optogenetics[28]. AAVs with Cre-dependent versions of the activating channelrhodopsin 2 (ChR2)-GFP or inhibitory soma-targeted *Guillardia theta* anion-conducting channelrhodopsins (stGtACR2)-FusionRed were injected into the ARC of male BNC2-Cre mice, after which an optical fibre was implanted over the ARC[28] (Fig. 3e and Extended Data Fig. 5b). Optogenetic activation of BNC2 neurons at a frequency of 20 Hz for 20 min led to a significant reduction in food intake in mice that were

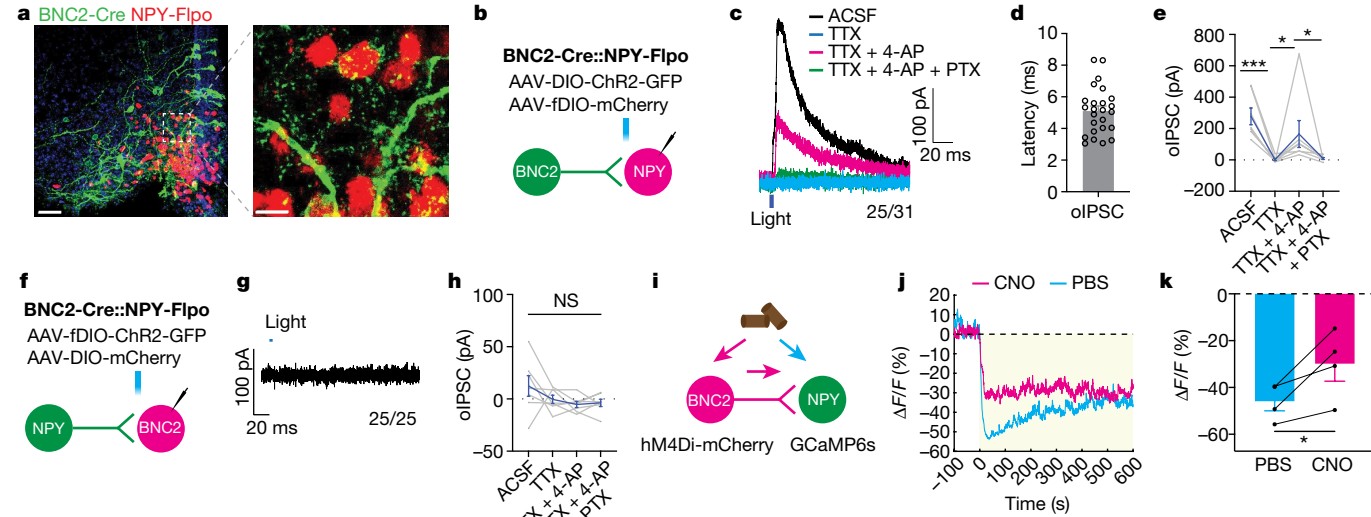

**Fig. 4 | BNC2 neurons monosynaptically inhibit AGRP/NPY neurons.**
**a**, Representative image of GFP and mCherry immunostaining in the ARC of adult BNC2-Cre::NPY-Flpo mice injected with AAV-DIO-ChR2-GFP and AAV-fDIO-mCherry ($n = 4$ mice). **b,f**, Schematics of ChR2-assisted circuity mapping to assess projections from BNC2 neurons to NPY neurons (**b**) or from NPY neurons to BNC2 neurons (**f**). **c**, Representative oIPSCs from NPY neurons following activation of BNC2 neurons with different blockers. **d**, oIPSC latency upon light stimulation ($n = 29$ cells from four mice). **e**, Quantification of oIPSC amplitude shown in **c** ($n = 7$ cells from four mice; Friedman test). **g**, Representative oIPSCs from BNC2 neurons following activation of NPY neurons. **h**, Quantification of oIPSC amplitudes under different conditions ($n = 7$ cells from four mice; two-way ANOVA). **i**, Experimental design and hypothesis. **j**, Average calcium traces from NPY neurons aligned to the time of chow presentation in mice receiving PBS (blue) or CNO (red). **k**, Quantification of fluorescence changes within the 0–200 s timeframe shown in **j** ($n = 4$ mice, paired Student's $t$-test). Data are presented as mean ± s.e.m. NS, not significant; *$P < 0.05$; ***$P < 0.001$. Statistical details in Source Data. Scale bars, 50 μm (**a**, left), 10 μm (**a**, right). ACSF, artificial cerebrospinal fluid.

fasted overnight compared with controls expressing GFP, and the decreased food intake persisted for up to 20 min after cessation of light stimulation (Fig. 3f). Activating ARC BNC2 neurons did not affect locomotion (Extended Data Fig. 5g). Consistent with the results from chemogenetic inhibition, optogenetic silencing of BNC2 neurons at 20 Hz significantly increased food intake in mice during the light cycle but this did not persist when the photoinhibition ceased (Fig. 3g).

We also used optogenetics to assess the effect of BNC2 neural activation on valence using a real-time place preference assay (Fig. 3h). We found a significant preference of mice for the chamber associated with BNC2 activation after an overnight fast compared with GFP expressing controls (Fig. 3i,j). However, there was no preference for the photostimulation-paired chamber in sated mice that were tested in the light cycle (Fig. 3j and Extended Data Fig. 5h).

## BNC2 neurons inhibit AGRP/NPY neurons

Food deprivation leads to the activation of AGRP/NPY neurons that is associated with negative valence, raising the possibility that BNC2 activation is associated with positive valence because they inhibit AGRP/NPY neurons[17,29]. We evaluated this using ChR2-assisted circuit mapping (CRACM) on brain slices[30]. A Flp-dependent mCherry AAV (fDIO-mCherry) and a Cre-dependent ChR2-GFP AAV (DIO-ChR2-GFP) were co-injected into the ARC of BNC2-Cre::NPY-FlpO mice[31]. We found dense GFP expression in BNC2 terminals within the ARC, in proximity to AGRP/NPY soma expressing mCherry (Fig. 4a). Evoked inhibitory postsynaptic currents (oIPSCs) were recorded in mCherry-labelled AGRP/NPY neurons before and after optogenetic activation of BNC2 neurons (Fig. 4b). We found synchronized oIPSCs after light activation in approximately 81% of AGRP/NPY neurons (25 of 31) with a rapid onset and a latency of 5.12 ms (Fig. 4c,d) indeed indicating that BNC2 neurons directly inhibit AGRP/NPY neurons. Consistent with this, the oIPSCs were abolished after the application of the sodium channel blocker tetrodotoxin (TTX), which blocks APs in all neurons, but inhibitory currents onto AGRP/NPY neurons were restored by the

co-application of 4-aminopyride (4-AP)—a potassium channel antagonist that enables ChR2-mediated depolarization of local presynaptic terminals (Fig. 4c,e). Finally, the oIPSCs observed in the presence of TTX and 4-AP were blocked after the application of the GABA$_A$ receptor antagonist picrotoxin (PTX) (Fig. 4c,e). In aggregate, these data show that BNC2 neurons directly inhibit AGRP/NPY neurons through the GABA$_A$ receptor.

We also tested whether AGRP/NPY neurons could inhibit BNC2 neurons by injecting AAV-fDIO-ChR2-GFP and AAV-DIO-mCherry into the ARC of BNC2-Cre::NPY-FlpO mice. However, in contrast to the effects of BNC2 activation to increase oIPSCs in AGRP/NPY neurons, we failed to see an effect of AGRP/NPY activation on postsynaptic currents in mCherry-labelled BNC2 neurons (0 of 25) (Fig. 4f–h). This was despite the fact that brief blue light pulses triggered APs in GFP-labelled AGRP/NPY neurons (Extended Data Fig. 6a,b). Thus, while BNC2 neurons directly inhibit AGRP/NPY neurons, AGRP/NPY neurons do not alter the activity of BNC2 neurons.

Previous reports have shown that AGRP/NPY neuronal activity is inhibited by the presence of food cues[17,29]. This raised the possibility that BNC2 neuronal activation might mediate some or all of the effect of food cues to inhibit AGRP/NPY neurons. We tested this by injecting AAV-DIO-hM4Di-mCherry, an inhibitory DREADD, together with an AAV-fDIO-GCaMP6s into the ARC of BNC2-Cre::NPY-FlpO (Fig. 4i). After an overnight fast, we administered PBS or CNO 30 min before providing the animal with chow (Extended Data Fig. 6c). Consistent with the previous results (Fig. 3), inhibition of BNC2 neurons significantly increased food intake relative to PBS-treated controls (Extended Data Fig. 6d). We then imaged the activity of AGRP/NPY neurons after refeeding using fibre photometry with or without inhibition of BNC2 neurons. As reported previously, control mice receiving PBS showed a robust reduction in calcium signals in AGRP/NPY neurons when given chow food (Fig. 4j,k). We then measured the activity of AGRP/NPY neurons after giving CNO and found that the inhibition of BNC2 neurons significantly attenuated the decrease in AGRP/NPY activity seen after refeeding (Fig. 4j,k). These findings indicate that a portion

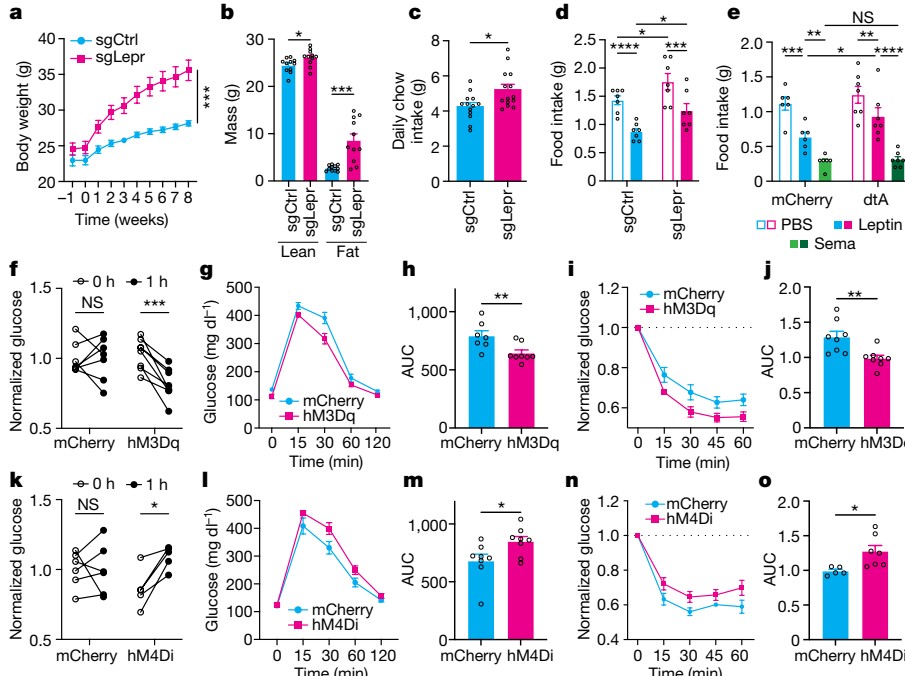

**Fig. 5 | LepR knockout in BNC2 neurons causes obesity. a**, Body weight of male mice on chow diet following sgCtrl or sgLepr injection (*n* = 14 mice per group; two-way ANOVA). **b**, Body composition of the two groups 8 weeks after virus injection (*n* = 11 mice per group, unpaired Student's *t*-test). **c**, Daily chow intake 8 weeks after injection (*n* = 13 sgCtrl-injected mice, *n* = 14 sgLepr-injected mice; Mann–Whitney test). **d**, Food intake 3 h after PBS or leptin injection at the onset of the night dark phase in male mice (*n* = 7 mice per group; two-way ANOVA). **e**, Food intake 3 h after PBS, leptin or Sema administration at the night phase onset in male mice (*n* = 6 mCherry-injected mice, *n* = 7 dtA-injected mice; two-way ANOVA). **f**, Glucose concentrations at 0 and 1 h after CNO injection following 16-h overnight fasting in male mice (*n* = 7 mCherry-injected mice, *n* = 8 hM3Dq-injected mice; paired Student's *t*-test or Wilcoxon test). **g,h**, GTT of two groups 1 h after CNO injection (*n* = 7 mCherry-injected mice, *n* = 8

hM3Dq-injected mice; Mann–Whitney test). GTT curves (**g**) and area under curve (AUC) (**h**). **i,j**, ITT of two groups of mice 1 h after CNO injection (*n* = 8 mCherry-injected mice, *n* = 8 hM3Dq-injected mice, unpaired Student's *t*-test). ITT curves (**i**) and AUC (**j**). **k**, Glucose concentrations at 0 and 1 h after CNO injection after 16-h overnight fasting in male mice (*n* = 6 mCherry-injected mice, *n* = 5 hM4Di-injected mice, paired Student's *t*-test). **l,m**, GTT 1 h after CNO injection (*n* = 8 mCherry-injected mice, *n* = 8 hM4Di-injected mice, unpaired Student's *t*-test). GTT curves (**l**) and AUC (**m**). **n,o**, ITT 1 h after CNO injection (*n* = 5 mCherry-injected mice, *n* = 7 hM4Di-injected mice, unpaired Student's *t*-test). ITT curves (**n**) and AUC (**o**). Data are presented as mean ± s.e.m. NS, not significant; \**P* < 0.05; \*\**P* < 0.01; \*\*\**P* < 0.001; \*\*\*\**P* < 0.001. Statistical details in Source Data.

of the sensory inputs that suppress AGRP/NPY neurons after refeeding are conveyed by BNC2 neurons.

## LepR knockout in BNC2 neurons causes obesity

BNC2 neurons express LepR (Fig. 1) and are activated by leptin (Fig. 2), indicating that they mediate some of leptin's effects. We assess this possibility by knocking out LepR in BNC2 neurons using CRISPR–Cas9 genome editing technology. BNC2-Cre mice were bred to LSL-Cas9-GFP mice[32], and an AAV carrying two single guide RNAs (sgRNAs) designed to target the mouse *Lepr* (sgLepr) locus, or two control guide RNAs (sgCtrl)[19,33], were injected bilaterally into the ARC. The specific deletion of the *Lepr* gene was confirmed by showing that leptin no longer increased the amounts of pSTAT3 in BNC2 neurons that received injections of the sgLepr guide RNAs compared with those receiving sgCtrl guide RNAs (Fig. 2 and Extended Data Fig. 7).

Male and female mice with a LepR knockout in BNC2 neurons gained significantly more weight compared with the sgCtrl-injected mice (Fig. 5a and Extended Data Fig. 8a) when fed on chow. Eight weeks after the guide RNA injection, fat mass within the sgLepr group was also significantly increased relative to the sgCtrl group with a small increase in lean mass (Fig. 5b and Extended Data Fig. 8b). Daily food intake was significantly higher in sgLepr mice at week 8, compared with their sgCtrl counterparts (Fig. 5c and Extended Data Fig. 8c). Total energy expenditure (EE) was increased significantly in the BNC2 LepR knockout mice, and this was still evident even when indexed to lean mass (Extended Data Fig. 8d–f). However, this difference was no longer significant

when the regression analysis of EE was calculated relative to body weight (Extended Data Fig. 8g). This indicates that the primary effect of the knockout to increase weight is a result of increased food intake. There was no difference in locomotor activity between the two groups (Extended Data Fig. 8h,i). The respiratory exchange ratio was higher in the sgLepr group in comparison with the sgCtrl group (Extended Data Fig. 8j,k), indicating that there is increased carbohydrate use. We next evaluated the response of the BNC2 LepR knockout mice to a highly palatable diet. BNC2 LepR male knockouts fed a standard high-fat diet (HFD, 60 kcal% fat) gained significantly more weight compared with sgCtrl-injected mice (Extended Data Fig. 8l). The sgLepr group also exhibited a substantial increase in fat mass, a small increase in lean mass and higher HFD consumption compared with controls (Extended Data Fig. 8m,n).

To further assess the role of BNC2 neurons in mediating leptin signalling, we examined how a LepR knockout in BNC2 neurons affects the response to leptin. In these studies, we assessed the response to leptin 2 weeks after virus injection, which was before the time that the BNC2 LepR knockout mice developed obesity. We administered PBS or leptin (5 mg kg⁻¹) to sgCtrl-injected or sgLepr-injected male mice immediately before the onset of the dark phase and measured food intake over the ensuing 3 h. Although leptin injection significantly reduced food intake in both groups, the sgLepr-injected mice consumed significantly more chow than did the sgCtrl-injected mice (Fig. 5d). These data show that LepR in BNC2 neurons is required for the full effect of leptin. In analogous studies, we ablated BNC2 neurons in the ARC and evaluated their response to leptin and glucagon-like peptide-1

(GLP-1). AAV-mCherry-flex-dtA was injected bilaterally into the ARC of adult male BNC2-Cre mice to ablate BNC2 neurons. Mice injected with AAV-DIO-mCherry served as controls. Here again, we found that the effect of leptin was decreased significantly in the dtA-injected mice versus the mCherry control mice receiving leptin (Fig. 5e). We also tested the effect of semaglutide (Sema)—a GLP-1 receptor agonist—in these animals. In contrast to the response to leptin, we found that Sema (10 nmol kg$^{-1}$) decreased food intake in both groups to a similar extent (Fig. 5e). This shows that, although BNC2 neurons are required for the complete response to leptin, they do not account for the effects of Sema.

Finally, we assayed glucose homeostasis in the BNC2 LepR knockouts and found that sgLepr male mice fed a chow diet for 9 weeks had increased fasting blood glucose levels (Extended Data Fig. 8o). The BNC2 LepR knockout mice also showed impaired glucose tolerance during a glucose tolerance test (GTT), and a significantly reduced response to insulin compared with sgCtrl in an insulin tolerance test (ITT) (Extended Data Fig. 8p–s). To evaluate whether the impaired glucose tolerance and insulin action in BNC2 LepR knockout mice was a direct effect or secondary to their obesity, we modulated the activity of BNC2 neurons using chemogenetics. AAVs expressing Cre-dependent activating or inhibitory DREADDs were injected into the ARC of BNC2-Cre male mice. Normal chow-fed mice receiving the activating hM3Dq DREADDs were fasted overnight, after which CNO was injected. At 1 h, the treated mice showed significantly decreased blood glucose levels compared with control mCherry-injected controls (Fig. 5f). We also observed an improvement in glucose tolerance and enhanced insulin sensitivity in the mice after activation of BNC2 neurons compared with control mice (Fig. 5g–j). Similar to the effects on valence, this effect is opposite to that seen after AGRP neuron activation[20]. Consistent with this, chemogenetic silencing of BNC2 neurons in chow-fed mice increased blood glucose levels and impaired glucose tolerance and insulin sensitivity (Fig. 5k–o). These data show that BNC2 neurons can acutely regulate peripheral glucose homeostasis independent of their effect on food intake and body weight.

## Discussion

Leptin regulates food intake and body weight by modulating the activity of discrete neural populations expressing LepR in the ARC and elsewhere[1], including orexigenic AGRP/NPY neurons and anorexigenic POMC neurons. However, several lines of evidence have raised the possibility that there are other, as yet unknown leptin-regulated neurons that suppress appetite[9,10,34]. However, the nature of this putative population(s) was not known.

Here we report the use of snRNA-seq to identify a new LepR-expressing neuron cluster in the ARC that expresses the *Bnc2* marker gene. Although there are fewer BNC2 neurons compared with those expressing POMC and AGRP/NPY, the amount of LepR in BNC2 neurons is as high or higher than in these other populations. The data further show that BNC2 neurons are activated rapidly by leptin or the presence of food and that the level of activation is further modulated by food palatability, nutritional status and food consumption (Extended Data Fig. 9). BNC2 neurons directly inhibit AGRP/NPY neurons and BNC2 activation reduces food intake for a sustained period. Activating BNC2 neurons also decreases peripheral glucose concentrations and increases insulin sensitivity while silencing them has the opposite effects. Finally, in contrast to anorexigenic POMC neurons, a BNC2-specific knockout of LepR causes significant hyperphagia and obesity. These findings establish a role for BNC2 neurons to acutely induce satiety in response to leptin and thus add an important new cellular component to the neural pathways that maintain homeostatic control of energy balance.

The *Bnc2* gene encodes a conserved transcription factor expressed during development and has been reported to play a role in cell development, proliferation, pigmentation and liver fibrosis[35–38]. In support of a potential role to regulate weight, genome-wide association studies have consistently linked the *Bnc2* gene to body mass index, fat distribution and diabetes[39,40], although the underlying mechanisms have not been explored previously. In the current paper, we used *Bnc2* as a marker gene, but these genetic data together with the data reported here also raises the possibility that *Bnc2* itself could have a role in the development or function of the BNC2/LepR neurons we identified. Further studies including knocking out the *Bnc2* gene exclusively in BNC2/LepR neurons will be necessary to assess this possibility and are now underway. While LepR clusters not expressing AGRP/NPY or POMC in the ARC have been identified previously in other studies, TRH, the canonical marker expressed in this cluster is also expressed in non-LepR clusters[22]. By contrast, BNC2 expression in the ARC is restricted to LepR neurons, making it particularly useful for probing the function of the neurons in this cluster. This enabled us to establish an important role for these neurons to regulate food intake and mediate leptin action. A recent report has indicated that TRH neurons in the ARC can also reduce food intake and body weight but the data indicate that these neurons are regulated by GLP-1, not leptin[41]. By contrast, as shown in this report, BNC2 neurons in ARC are regulated by leptin but not GLP-1, indicating that BNC2 and TRH neurons in ARC are functionally distinct.

The canonical 'yin–yang' model of feeding regulation has emphasized the opposing roles of POMC neurons (suppressing hunger) and AGRP/NPY neurons (promoting hunger)[2]. However, as described above, the characteristics of these neural populations are markedly different in several respects, indicating that there might be a missing leptin-regulated neuronal population that acutely suppresses food intake. Consistent with this, a knockout of LepR in GABAergic neurons had a greater effect than did a knockout of LepR in AGRP or POMC neurons[9]. The identity of this 'missing' GABAergic population was not known and the evidence reported here showing that BNC2/LepR neurons rapidly suppress feeding indicates that they serve this function. However, these data do not exclude the possibility that other populations might also contribute.

Optogenetic activation of BNC2 neurons potently reduces food intake for a sustained period even after photoactivation has ceased. This is analogous to the sustained effect of AGRP/NPY neuronal activation to increase in food intake through the release of neuropeptide Y[42,43]. The sustained satiety effect of activating BNC2 neurons lasted at least 20 min, possibly longer. Previous research indicates that sustained hunger after AGRP/NPY neuron activation can extend up to 1 h (ref. 42). BNC2 neurons are GABAergic and directly inhibit AGRP/NPY neurons through the GABA$_A$ receptor although, as is the case for other feeding neurons, it is possible that neuropeptides also contribute. The snRNA-seq data shows expression of several putative neuropeptides in BNC2 neurons including *Scg2*, *Chga* and *Trh*, which have been implicated in feeding control and energy balance[44–47]. Future studies will be necessary to determine whether, in addition to GABA, these neuropeptides mediate some of the effects of BNC2 neurons.

BNC2 neurons suppress appetite, at least in part, by directly inhibiting AGRP/NPY neurons through monosynaptic connections. However, as is the case for POMC neurons, this interaction is not reciprocal[48,49] as AGRP/NPY neurons do not seem to modulate the activity of BNC2 neurons. Similar to AGRP/NPY neurons, BNC2 neurons are activated by sensory cues associated with food and may function as coincidence detectors for both sensory and interoceptive inputs such as leptin. The rapid regulation of AGRP/NPY and BNC2 neurons by food cues is probably mediated by associative sensory inputs, the precise source of which is unknown at present. It is plausible that these sensory inputs synapse onto both BNC2 neurons and AGRP/NPY neurons, exerting distinct effects on these two neuronal populations. If true, mapping the afferent inputs to both may reveal anatomic sites that convey associative sensory inputs. Alternatively, the food cues may target primarily BNC2 neurons, which subsequently relay the information to AGRP/NPY neurons. In support of the former, we find that inhibiting BNC2 neurons only partially blunts the effect of food cues to suppress

AGRP/NPY activity, indicating that both direct and indirect (through BNC2 neurons) pathways contribute. It has also been shown that while food cues can transiently suppress AGRP/NPY activity, caloric content and consumption are required for a sustained reduction in their activity[50]. This is also the case for BNC2 neurons because the level of BNC2 neuron activity decreases markedly when the food is removed.

In addition to markedly increasing weight, deletion of LepR in adult BNC2 neurons resulted in abnormal GTTs and ITTs, indicating a potential role of BNC2 neurons to also regulate glucose metabolism. Indeed, activating BNC2 neurons in the ARC decreases peripheral glucose concentrations and enhances insulin sensitivity, whereas inhibiting them has the opposite effect, extending their role beyond feeding regulation. Here again, some of these effects could be a result of the effect of BNC2 neurons on AGRP/NPY neural activity.

In summary, we have identified BNC2 neurons as a fast-acting population of neurons that acutely regulate feeding and energy balance bidirectionally. These findings add an important new component to the neural circuit that regulates appetite and adiposity, while also shedding new light on the mechanisms by which leptin regulates this system. Finally, pharmacologic activation of these neurons could have therapeutic implications to reduce weight or suppress the negative valence associated with hunger.

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

## Methods

### Mice

All animal experiments were approved by the Institutional Animal Care and Use Committee at Rockefeller University and were carried out in accordance with the National Institutes of Health guidelines. Mice were group housed in a 12-h light/12-h dark cycle at 22 °C and 30–60% humidity with ad libitum access to a normal chow diet and water. We used the following mouse genotypes: C57BL/6J (wild type (WT); stock no. 000664, The Jackson Laboratory), NPY-IRES2-FlpO-D (The Jackson Laboratory, stock no. 030211), Rosa26-LSL-Cas9 (The Jackson Laboratory, stock no. 024857) and BNC2-P2A-iCre (see below). For all Cre or Flp mouse line experiments, only heterozygous animals were used. Sample sizes were decided on the basis of experiments from similar studies. Littermates of the same sex were assigned randomly to either experimental or control groups. The experiments were performed using both male and female mice, as indicated.

### Generation of BNC2-P2A-iCre mouse line

The BNC2-P2A-iCre mouse line was generated by the CRISPR and Genome Editing Resource Center and Transgenic and Reproductive Technology Resource Center at Rockefeller University using CRISPR–Cas9 technology[51]. Briefly, a custom-designed long single-stranded DNA (lssDNA) donor, containing homology arms of *Bnc2* locus flanking the P2A-iCre sequence, was inserted near the endogenous STOP codon. Two guide RNAs (sgRNAs) were used to induce site-specific double-stranded breaks. The lssDNA donor with the pre-assembled Cas9 protein–gRNA complexes was mixed and microinjected into C57BL/6J mouse zygotes following standard CRISPR genome engineering protocols. The resulting live offspring were genotyped by PCR with two sets of primers that specifically amplified the mutant allele. Validation was ensured by Sanger sequencing. The BNC2-P2A-iCre transgenic animals were bred to C57BL/6J mice for maintenance.

### Viruses

AAVs used in these studies were obtained from Addgene, UNC Vector Core, or generated by Janelia Viral Tools Service. We used the following viruses: AAV5-hSyn-DIO-hM3D(Gq)-mCherry (Addgene, catalogue no. 44361, $2.2 \times 10^{13}$ vg ml$^{-1}$), AAV5-hSyn-DIO-hM4Di(Gi)-mCherry (Addgene, catalogue no. 44362, $2.5 \times 10^{13}$ vg ml$^{-1}$), AAV5-hSyn-DIO-mCherry (Addgene, catalogue no. 50459, $2.2 \times 10^{13}$ vg ml$^{-1}$), AAV5-Ef1a-DIO-EYFP (Addgene, catalogue no. 27056, $1.6 \times 10^{13}$ vg ml$^{-1}$), AAV5-hSyn-Flex-GCaMP6s-WPRE (Addgene, catalogue no. 100845, $2.9 \times 10^{13}$ vg ml$^{-1}$), AAV5-EF1a-DIO-hChR2(H134R)-EYFP (UNC Vector Core, $2.7 \times 10^{12}$ vg ml$^{-1}$), AAV1-hSyn1-SIO-stGtACR2-FusionRed (Addgene, catalogue no. 105677, $2.1 \times 10^{13}$ vg ml$^{-1}$), AAV5-Ef1a-fDIO-mCherry (Addgene, catalogue no. 114471, $2.3 \times 10^{13}$ vg ml$^{-1}$), AAV8-Ef1a-fDIO-GCaMP6s (Addgene, catalogue no. 105714, $2.1 \times 10^{13}$ vg ml$^{-1}$), AAV5&DJ-EF1a-fDIO-hChR2(H134R)-EYFP-WPRE (UNC Vector Core, $1.4 \times 10^{12}$ vg ml$^{-1}$), AAV5-Ef1a-mCherry-flex-dtA (Addgene, catalogue no. 58536, $3.88 \times 10^{12}$ vg ml$^{-1}$). For *Lepr* deletion, AAV viral vectors were cloned inhouse and packaged with the AAV5 serotype using Janelia Viral Tools Service. The sequences of sgLepR are: 5′-GAGTCATCGGTTGTGTTCGG-3′, 5′-TGCCGGCGGTTGGATGGACT-3′ (virus titre, $4.9 \times 10^{12}$ vg ml$^{-1}$); The sequence of sgCtrl is: 5′-TTTTTTTTTTTTTTTGAATTC-3′ (virus titre, $8.5 \times 10^{12}$ vg ml$^{-1}$). Viral aliquots were stored at −80 °C before stereotaxic injection.

### Chemicals and diet

The following chemicals were used in this study: Leptin (ThermoFisher Scientific, catalogue no. 498OB05M, 3 mg kg$^{-1}$ or 5 mg kg$^{-1}$, intraperitoneal injection), Sema (Millipore Sigma, catalogue no. AT35750, 10 nmol kg$^{-1}$, subcutaneous injection), CNO dihydrochloride (Tocris, catalogue no. 6329, 3 mg kg$^{-1}$, intraperitoneal injection), sucrose tablets 20 mg (Test-Diet, catalogue no. 1811555) and HFD (Research Diets, 60% kcal% Fat).

### Stereotactic surgery

Mice (8–10 weeks old) were anaesthetized using isoflurane anaesthesia (induction 5%, maintenance 1.5–2%) and positioned on a stereotaxic rig (Kopf Instruments, Model 1900). Viruses were delivered into the brains through a glass capillary using a Drummond Scientific Nanoject III Programmable Nanoliter Injector. For the ARC region, the following coordinates relative to the bregma were used: anterior–posterior, −1.65 mm to −1.70 mm; medial–lateral (ML), ±0.25 mm to 0.30 mm and dorsal–ventral (DV), −5.9 mm. For chemogenetics experiments, Bnc2 neuron labeling and *Lepr* deletion, 30–50 nl of the virus was injected bilaterally at a rate of 1 nl s$^{-1}$. For optogenetics, 30 nl of the virus was injected unilaterally at a rate of 1 nl s$^{-1}$ followed by the implant of an optical fibre (ThorLabs, catalogue no. CFM12U-20) at 200 μm above the ARC (anterior–posterior, −1.65 mm; ML, 0.3 mm; DV, −5.7 mm). For fibre photometry experiments, 30 nl of the virus was injected unilaterally followed by the implant of an optical fibre cannula (Doric, catalogue no. MFP_400/430/1100-0.57_1m_FCM-MF2.5_LAF) at 150 μm above the ARC (anterior–posterior, −1.65 mm; ML, 0.3 mm; DV, −5.75 mm). For CRACM experiments, the two viruses were mixed at the ratio of 1:1, and 50 nl of the mixed virus was injected bilaterally into the ARC.

### Isolation of nuclei and snRNA-seq

Male C57BL/6J mice aged 10–12 weeks were euthanized by transcardial perfusion using ice-cold HEPES-Sucrose Cutting Solution containing NaCl (110 mM), HEPES (10 mM), glucose (25 mM), sucrose (75 mM), MgCl$_2$ (7.5 mM) and KCl (2.5 mM) at pH 7.4 (ref. 52). Subsequently, brains were dissected quickly in the same solution, frozen using liquid nitrogen and stored at −80 °C until nuclei isolation. To isolate nuclei, as described previously[53,54], the samples were thawed on ice, resuspended in HD buffer containing tricine KOH (10 mM), KCl (25 mM), MgCl$_2$ (5 mM), sucrose (250 mM), 0.1% Triton X-100, SuperRNaseIn (0.5 U ml$^{-1}$), RNase Inhibitor (0.5 U ml$^{-1}$). Samples were homogenized using a 1 ml dounce homogenizer. The resulting homogenates were filtered using a 40 μM filter, centrifuged at 600$g$ for 10 min and resuspended in nucleus storage buffer containing sucrose (166.5 mM), MgCl$_2$ (10 mM), Tris buffer (pH 8.0, 10 mM), SuperRNaseIn (0.05 U ml$^{-1}$), RNase Inhibitor (0.05 U ml$^{-1}$) for subsequent staining. Nucleus quality and number were assessed using an automated cell counter (Countess II, ThermoFisher). For staining, nuclei were labelled with Hoechst 33342 (ThermoFisher Scientific, catalogue no. H3570; 0.5 μl per million nuclei), anti-NeuN Alexa Fluor 647-conjugated antibody (Abcam, catalogue no. ab190565; 0.5 μl per million nuclei) and TotalSeq anti-Nuclear Pore Complex Proteins Hashtag antibody (BioLegend, catalogue no. 682205; 0.5 mg per million nuclei) for 15 min at 4 °C. Following staining, samples were washed with 10 ml 2% BSA (in PBS) and centrifuged at 600$g$ for 5 min. Nuclei were then resuspended in 2% BSA (in PBS) with RNase inhibitors (SuperRNaseIn 0.5 U ml$^{-1}$, RNase Inhibitor 0.5 U ml$^{-1}$) for subsequent fluorescence-activated cell sorting. The samples were gated on the basis of Hoechst fluorescence to identify nuclei and then further sorted on the basis of high Alexa Fluor 647 expression, designating NeuN$^+$ nuclei as neurons.

Nuclei were captured and barcoded using 10x Genomics Chromium v.3 following the manufacturer's protocol. The processing and library preparation were carried out by the Genomics Resource Center at Rockefeller University, and sequencing was performed by Genewiz using Illumina sequencers.

### SnRNA-seq analysis

The FASTQ file was analysed with Cell Ranger v.5.0. The snRNA-seq data for ARC (WT) was preprocessed individually using the Seurat v.4 (v.4.0.3)[55]. Cells with more than 800 and fewer than 5,000 RNA features were selected for further analysis. Cells with greater than 1% mitochondrial genes and greater than 12,000 total RNA counts were also removed. Genes detected in fewer than three cells were excluded. We then demultiplexed the cells on the basis of their hashtag count

(positive quantile = 0.8) using the built-in function in Seurat v.4. Only the cells with singlet Hashtag assignment were kept for downstream analysis. The data was then log-normalized with a scale factor of 10,000. After the initial quality control, demultiplexing and normalization steps, all the singlets were then scaled and reduced dimensionally with principal component analysis and uniform manifold approximation and projection (UMAP). Leiden clustering (resolution = 0.55) was applied to identify clusters. We used known cell-type specific gene expression to annotate the clusters.

We analysed co-expression of marker genes within the human ARC using previously published human adult samples, and the data can be accessed through the NeMO archive (https://assets.nemoarchive.org/dat-917e9vs). A cell was considered to express the marker gene if at least two unique molecular identifiers were detected. The identification of arcuate cells was achieved by clustering and the expression of canonical markers, as detailed in the earlier study. Co-expression of genes such as *Lepr*, *Bnc2*, *Agrp*, *Npy* and *Pomc* was tabulated in R, and two-tailed Fisher's tests were calculated to assess the significance of co-expression of gene pairs within the 16,819 arcuate cells in the human dataset.

### Chemogenetics for activation or inhibition
AAV viruses were delivered bilaterally into the ARC of male BNC2-Cre mice aged 8–10 weeks. Mice were then allowed to recover and express DREADDs for at least 3 weeks. For activation or inhibition, animals were injected intraperitoneally with 3 mg kg$^{-1}$ of CNO or PBS (control).

### Optogenetics for activation or inhibition
AAV viruses were delivered unilaterally into the ARC of male BNC2-Cre mice aged 8–10 weeks followed by the implantation of an optic fibre. Subsequently, the mice were given a recovery period of at least 3 weeks to allow for gene expression. Before the experiments, the mice were habituated to patch cables over a period of 5 days. The implanted optic fibres were connected to patch cables using ceramic sleeves (Thorlabs) and linked to a 473 nm laser (OEM Lasers/OptoEngine). The output of the laser was verified at the beginning of each experiment. A blue light, generated by a 473 nm laser diode (OEM Lasers/OptoEngine) with a power of 15 mW, was used. The light pulse (10 ms) and frequency (20 Hz) were controlled by a waveform generator (Keysight) to either activate or inhibit BNC2 neurons in the ARC. In the activation feeding experiments, mice were allowed to acclimate to the cage for 20 min. Subsequently, three feeding sessions, each lasting 20 min, were initiated. During these sessions, the light was turned off for the initial 20 min, switched on for the subsequent 20 min and then turned off again for the remaining 20 min. In the inhibition feeding experiments, following the 20 min acclimation, each feeding session was extended to 30 min. The amount of food consumed during each feeding session was recorded manually. Animals were euthanized at the end of the experiments to confirm viral expression and fibre placement using immunohistochemistry.

### Real-time place preference
A custom-made two-chamber box (50 × 50 × 25 cm black plexiglass) with an 8.5 cm gap enabling animals to move freely between the chambers was used for this assay. To evaluate the initial preference of the mice, they were introduced into the box for a 10 min session without any photostimulation. Subsequently, in the second 10 min session following the initial one, photostimulation (15 mW, 20 Hz) was paired with the chamber for which the mice exhibited less preference during the initial session. The Ethovision XT v.13 software, coupled with a CCD camera, facilitated the recording of the percentage of time spent by the mice in each chamber.

### Fibre photometry
Mice were acclimated to tethering and a home-cage-style arena for 5 min daily over the course of 5 days before the experiment. Data acquisition was conducted using a fibre photometry system from Tucker-Davis Technologies (catalogue no. RZ5P, Synapse) and Doric components, with recordings synchronized to video data in Ethovision by transistor–transistor logic triggering. A dual fluorescence Mini Cube (Doric) combined light from 465 nm and isosbestic 405 nm light-emitting diodes (LEDs), which were transmitted through the recording fibre connected to the implant. GCaMP6s fluorescence, representing the calcium-dependent signal (525 nm), and isosbestic control (430 nm) were detected using femtowatt photoreceivers (Newport, catalogue no. 2151) and a lock-in amplifier at a sampling rate of 1 kHz. Analysis was conducted using a Matlab script involving the removal of bleaching and movement artifacts using a polynomial least square fit applied to the 405 nm signal, adjusting it to the 465 nm trace ($405_{fitted}$), and then calculating the GCaMP signal as %$\Delta F/F = (465_{signal} − 405_{fitted})/405_{fitted}$. The resulting traces were filtered using a moving average filter and down-sampled by a factor of 20. The code is available upon request.

### In situ hybridization
Mice were briefly transcardially perfused with ice-cold RNase-free PBS. Brains were then quickly collected, embedded in optimal cutting temperature embedding medium on dry ice, and stored at −80 °C until cryostat sectioning (15 μm thickness) onto Superfrost Plus Adhesion Slides (ThermoFisher). The RNAscope Fluorescent Multiplex assay (Advanced Cell Diagnostics Bio) was based on the manufacturer's protocol. All reagents were purchased from Advanced Cell Diagnostics (ACDbio). Probes for the following mRNAs were used: *Agrp* (catalogue no. 400711-C3), *Pomc* (catalogue no. 314081-C3), *Lepr* (catalogue no. 402731), *Slc31a1* (catalogue no. 319191) and *Bnc2* (catalogue no. 518521-C2). Briefly, brain sections were fixed in 4% paraformaldehyde (PFA) at 4 °C for 15 min followed by serial submersion in 50% ethanol, 70% ethanol, and twice in 100% ethanol for 5 min each at room temperature. Sections were treated with Protease IV for 30 min at room temperature followed by a 2 h incubation with specific probes at 40 °C using a HyBez oven. Signal amplification was achieved through successive incubations with Amp-1, Amp-2, Amp-3 and Amp-4 for 30, 15, 30 and 15 min, respectively, at 40 °C using a HyBez oven. Each incubation step was followed by two 2 min washes in RNAscope washing buffer. Nucleic acids were counterstained with DAPI Fluoromount-G (SouthernBiotech) mounting medium before coverslipping. The slides were visualized using an inverted Zeiss LSM 780 laser scanning confocal microscope with a ×20 lens. The acquired images were imported into Fiji for further analysis.

### Immunohistochemistry
Mice were perfused transcardially with PBS first and then 4% PFA for fixation. Brains were collected and immersed in 4% PFA overnight at 4 °C for more fixation. Fixed brains were immersed sequentially in 10% sucrose, 20% sucrose and 30% sucrose buffers for 1 h, 1 h and overnight, respectively, all at 4 °C. After this, the brains were embedded in optimal cutting temperature embedding medium and stored at −80 °C until cryostat sectioning (30–50 μm thickness). For the staining process, brain sections were first blocked in a blocking buffer containing 3% BSA, 2% goat serum and 0.1% Triton X-100 in PBS for 30 min at room temperature followed by an overnight incubation with primary antibodies in the cold room. After washing in PBS, the sections were incubated with fluorescence-conjugated secondary antibodies (Invitrogen) for 1 h at room temperature. Stained sections were mounted onto SuperFrost (Fisher Scientific catalogue no. 22-034-980) slides and then visualized with an inverted Zeiss LSM 780 laser scanning confocal microscope with a ×10 or ×20 lens. The acquired images were imported to Fiji for further analysis. The following antibodies were used: FOS antibody (1:1,000; Synaptic systems, catalogue no. 226308), pSTAT3 antibody (1:1,000; Cell Signaling Technology, catalogue no. 9145 s), GFP (1:1,000; abcam, catalogue no. ab13970), RFP (1:1,000; Rockland, catalogue no. 600-401-379).

### Electrophysiology and CRACM

Adult mice were euthanized by transcardial perfusion using ice-cold cutting solution containing choline chloride (110 mM), $NaHCO_3$ (25 mM), KCl (2.5 mM), $MgCl_2$ (7 mM), $CaCl_2$ (0.5 mM), $NaH_2PO_4$ (1.25 mM), glucose (25 mM), ascorbic acid (11.6 mM) and pyruvic acid (3.1 mM). Subsequently, brains were quickly dissected in the same solution and sectioned using a vibratome into 275 μm coronal sections. These sections were then incubated in artificial cerebral spinal fluid containing NaCl (125 mM), KCl (2.5 mM), $NaH_2PO_4$ (1.25 mM), $NaHCO_3$ (25 mM), $MgCl_2$ (1 mM), $CaCl_2$ (2 mM) and glucose (11 mM) at 34 °C for 30 min, followed by room temperature incubation until use. The intracellular solution for current-clamp recordings contained K-gluconate (145 mM), $MgCl_2$ (2 mM), $Na_2ATP$ (2 mM), HEPES (10 mM) and EGTA (0.2 mM, 286 mOsm, pH 7.2). The intracellular solution for the voltage-clamp recording contained $CsMeSO_3$ (135 mM), HEPES (10 mM), EGTA (1 mM), QX-314 (chloride salt, 3.3 mM), Mg-ATP (4 mM), Na-GTP (0.3 mM) and sodium phosphocreatine (8 mM, pH 7.3 adjusted with CsOH). Signals were acquired using the MultiClamp 700B amplifier and digitized at 20 kHz using DigiData1550B (Molecular Devices). The recorded electrophysiological data were analysed using Clampfit (Molecular Devices) and MATLAB (Mathworks).

For CRACM experiments, voltage-clamp recordings were conducted on BNC2 and NPY neurons. To record oIPSCs, the cell membrane potential was held at 0 mV. ChR2-expressing axons were activated using brief pulses of full-field illumination (0.5 ms, 0.1 Hz, ten times) onto the recorded cell with a blue LED light (pE-300 white; CoolLED). Subsequently, TTX (1 μM), 4-AP (100 mM) and PTX (1 μM) were applied sequentially through the bath solution, each for 10–20 min. Data acquisition started at least 5 min after each drug application.

### Indirect calorimetry

Indirect calorimetry was performed using the Phenomaster automated home cage phenotyping system (TSE Systems). Mice were housed individually in environmentally controlled chambers maintained at 22 °C, following a 12 h light/12 h dark cycle, and at 40% humidity, with ad libitum access to food and water. $O_2$ and $CO_2$ measurements were collected at 15 min intervals with a settling time of 3 min and a sample flow rate of 0.25 l min$^{-1}$. The raw data obtained were analysed using CalR[56].

### Blood glucose, GTT and ITT

Blood glucose levels were measured using a OneTouch Ultra meter and glucose test strips. For GTTs, mice were fasted overnight followed by a 20% glucose injection (2 g kg$^{-1}$) and glucose measurements at 0, 15, 30, 60 and 120 min. ITTs were conducted after a 4 h fast, with insulin injection (0.75 U kg$^{-1}$) and glucose measurements at 0, 15, 30, 45 and 60 min. To test how BNC2 neuron activity affected glucose metabolism, CNO was injected for 1 h before the start of GTT and ITT experiments.

### Statistical analysis

All statistical analyses used GraphPad Prism v.9. Data distribution was tested for normality (Shapiro–Wilk test) and then comparisons were made using parametric or non-parametric tests, as appropriate. Two-tailed statistical tests were used, and statistical significance was determined by Student's $t$-test, Mann–Whitney test, Fisher's exact test, one-way or 2-way ANOVA, and Friedman test as indicated in the Source Data.

### Reporting summary

Further information on research design is available in the Nature Portfolio Reporting Summary linked to this article.

## Data availability

All data generated or analysed during this study are included in the manuscript and supporting files. The raw sequencing data of adult mouse ARC have been deposited in the Gene Expression Omnibus and are available under the accession number GSE249564. The snRNA-seq data for adult human ARC are accessible through the NeMO archive (https://assets.nemoarchive.org/dat-917e9vs). Source Data are provided with this paper.

## Code availability

The analytic code for processing data is available at GitHub (https://github.com/yuqiyuqitan/BNC2).

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

**Acknowledgements** We thank all members of the laboratory of J.M.F. for discussions and support, H. Duan from the Genomics Resource Center and Flow Cytometry Resource Center for technical assistance with snRNA-seq experiments, C. Yang from CRISPR and Genome Editing Center for generating the BNC2-P2A-iCre mouse line, and the Bio-Imaging Resource Center. This work was supported by the JPB Foundation and the Howard Hughes Medical Institute (to J.M.F.), the Kavli Neural Systems Institute Postdoctoral Fellowship (to H.L.T.), and the Maurice R. and Corinne P. Greenberg Center for the Advancement of Translational Research Pilot Project Grant (to H.L.T.). Figures 2h,n and 3e,h and Extended Data Figs. 2a,c and 6c were created with Biorender (https://biorender.com).

**Author contributions** H.L.T. conceived the study, designed the experiments and performed all experiments with input from all authors. H.L.T. also led the analysis of the research data. L.Y. conducted the electrophysiology experiments and analysed the data. Y.T. and P.W. analysed the mouse snRNA-seq data. A.I. performed the mouse snRNA-seq experiments. K.P. performed indirect calorimetry experiments. J.I. performed the immunohistochemistry experiments. B.R.H. analysed the human snRNA-seq data. P.C. and D.L. provided inputs on experimental design. C.K. provided the code for fibre photometry data analysis. H.L.T. and J.M.F. wrote the manuscript.

**Competing interests** J.M.F. receives royalty payments from the sale of leptin through Rockefeller University, according to the university's policy for distributing proceeds from inventions to their inventors. All other authors declare no competing interests.

**Additional information**
**Correspondence and requests for materials** should be addressed to Jeffrey M. Friedman.

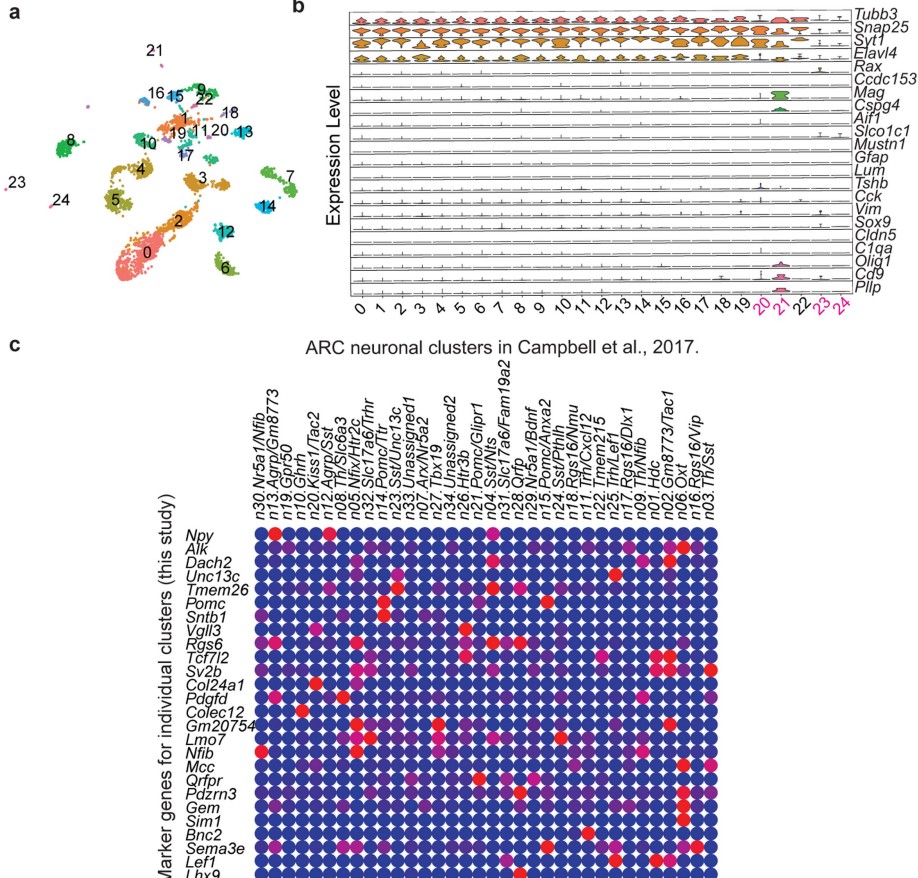

**Extended Data Fig. 1 | SnRNA-seq analysis of mouse ARC neurons.**
**a**, Single-cell UMAP plot of the male ARC (N = 6 adult WT mice). **b**, Violin plot showing known markers of different cell types: neurons (*Tubb3*, *Snap25*, *Syt1*, *Elavl4*), tanycytes (*Rax*), ependymocytes (*Ccdc153*), oligodendrocyte cells (*Mag*, *Olig1*, *Cspg4*, *Cd9*, *Pllp*), macrophages (*Aif1*, *C1qa*), endothelial cells (*Slco1c1*, *Cldn5*), mural cells (*Mustn1*), astrocyte (*Gfap*, *Sox9*), vascular and leptomeningeal cells (*Lum*), pituitary cells (*Tshb*), pars tuberalis cells (*Cck*), ependymal cells (*Vim*). Clusters 20/21/23/24 are non-neuronal clusters. **c**, Dot plot displaying the expression of cluster-specific marker genes (identified in this study) across neuronal clusters with the ARC from Campbell et al.[22].

**a**

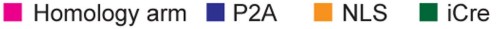

ATG TGA

BNC2-P2A-iCre

P2A iCre

Protein BNC2 iCre

**b**

gRNAs used for gene editing:
gRNA1: GGACTAATCTATTGAAGTGAAGG
gRNA2: TCAATAGATTAGTCCCAGAACGG

**lssDNA template sequence:**

acctgaatgggtacgggagaggcatggcagaggactacatggtccttgacctgagtaccacctccagcctccagtccagcagcagtgtccatt
cctccagagagtccgatgcaggcagcgatgaggggattcttctcgacgacatcgatggcgcgagtgacagtggggaatccactcacaaggc
cgaggcccccaccctccccggcagcctcggggctgaagtttcaggatctcttatgttcagcagtttgtctgggagcaatggtgggatcatgtgca
acatttgccacaaaatgtacagcaacaaggggaccctgcgggtccactacaaaactgtgcatttgcgagagatgcataagtgcaaagtcccc
ggctgtaacatgatgttctcctctgtgcgaagccgaaacaggcacagccagaaccccaatcttcacaaaaacattccctttacttcaatagatgg
aagcggagctactaacttcagcctgctgaagcaggctggagacgtggaggagaaccctggacctatggtgcccaagaagaagaggaaag
tctccaacctgctgactgtgcaccaaaacctgcctgccctccctgtggatgccacctctgatgaagtcaggaagaacctgatggacatgttcagg
gacaggcaggccttctctgaacacacctggaagatgctcctgtctgtgtgcagatcctgggctgcctggtgcaagctgaacaacaggaaatgg
ttccctgctgaacctgaggatgtgagggactacctcctgtacctgcaagccagaggcctggctgtgaagaccatccaacagcacctgggcca
gctcaacatgctgcacaggagatctggcctgcctcgcccttctgactccaatgctgtgtccctggtgatgaggagaatcagaaaggagaatgtg
gatgctggggagagagccaagcaggccctggcctttgaacgcactgactttgaccaagtcagatccctgatggagaactctgacagatgcca
ggacatcaggaacctggccttcctgggcattgcctacaacaccctgctgcgcattgccgaaattgccagaatcagagtgaaggacatctcccg
caccgatggtgggagaatgctgatccacattggcaggaccaagaccctggtgtccacagctggtgtggagaaggccctgtccctgggggtta
ccaagctggtggagagatggatctctgtgtctggtgtggctgatgacccccaacaactacctgttctgccgggtcagaaagaatggtgtggctgcc
ccttctgccacctcccaactgtccacccgggccctggaagggatctttgaggccacccaccgcctgatctatggtgccaaggatgactctgggc
agagatacctggcctggtctggccactctgccagagtgggtgctgccagggacatggccagggctggtgtgtccatccctgaaatcatgcagg
ctggtggctggaccaatgtgaacattgtgatgaactacatcagaaacctggactctgagactggggccatggtgaggctgctcgaggatgggg
actgatcccagaacggacactccaaatgccagctctcaccagatggcctacatgtttgaactgccatagccagtgtgtgcttccgcagtggggc
acatatgtgtgtgtacatatgtgtgcctctgtgtctacacttgtgcacacacatactttctttctttagataaaaatgataaatactaggtgctttgaaat
tttttcttttcctttatagttttgggagagggtggatctttacttggggtaaaaacaagagtgcccttttcagcacacacacacacacacacacaca
cacacaaacacaacataacacacaaagtgtgcaactaaccccaattttgataggataattcttggtcttctccaaagagacattttgttgtacctat
gactgttgcctgcaaaaacaataaataaat.aaaaggggaaaaaagagaaaagaaacaaaaaggaacttcttacagttgtct

■ Homology arm ■ P2A ■ NLS ■ iCre

**c**

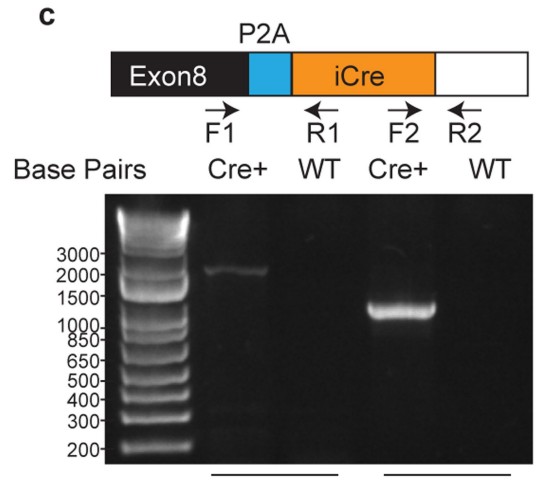

P2A

Exon8 iCre

F1 → ← R1 F2 → ← R2

Base Pairs Cre+ WT Cre+ WT

3000
2000
1500
1000
850
650
500
400
300
200

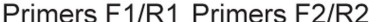

Primers F1/R1 Primers F2/R2

**d**

Bnc2/mCherry

Bnc2

mCherry

**Extended Data Fig. 2 | Generation of BNC2-P2A-iCre knockin mouse line.**
**a**, Schematic of the BNC2-P2A-iCre locus. **b**, Sequence information of gRNAs
and lssDNA. **c**, Genomic PCR using specific primers (N = 3 mice). **d**, Cre expression
was assessed using a Cre-dependent mCherry viral construct. Colocalization
with endogenous *Bnc2* mRNA was confirmed through RNA ISH (N = 3 mice).
Scale bar, 20 μm. Illustrations in **a** and **c** were created using BioRender (https://
biorender.com).

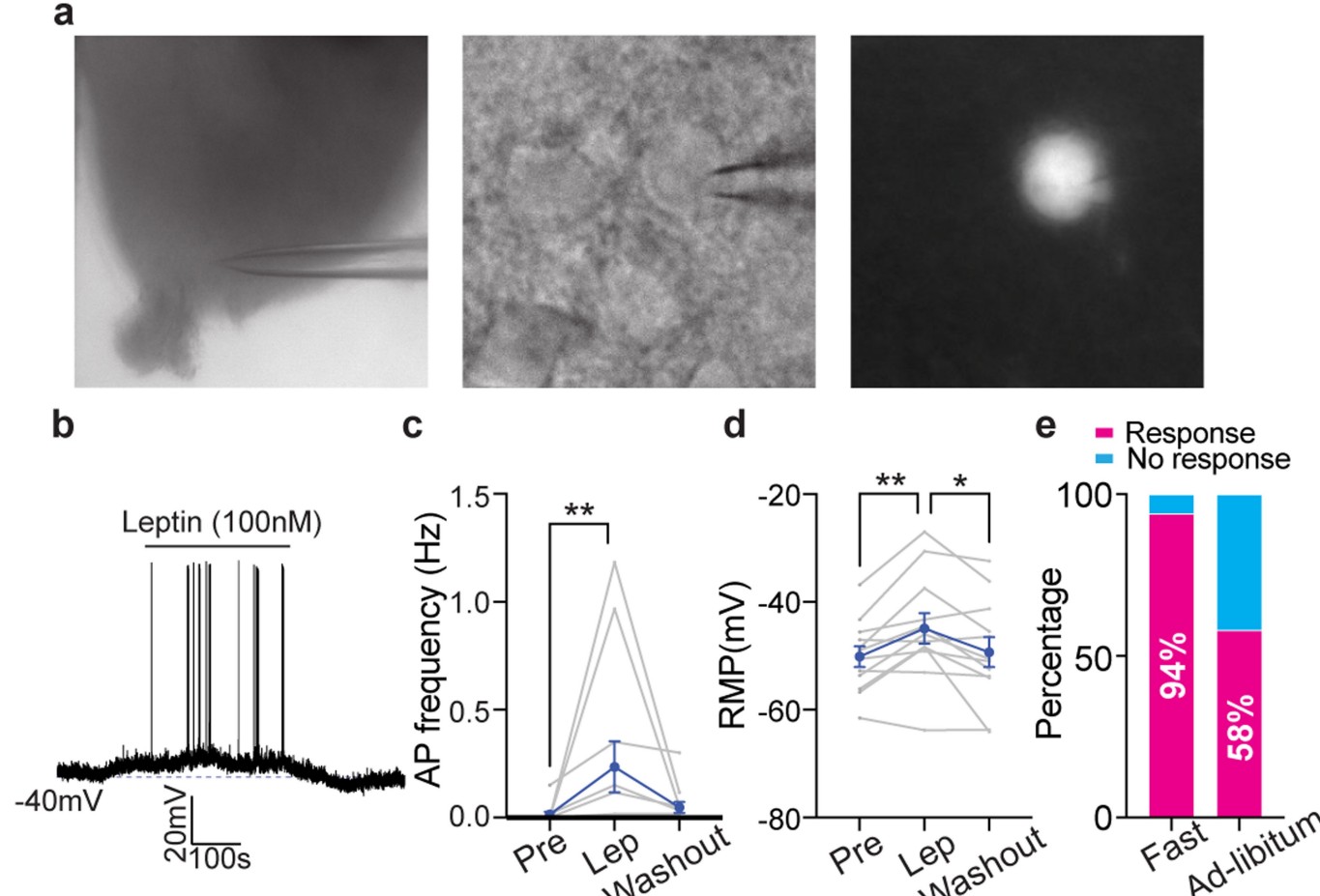

**Extended Data Fig. 3 | Activation of BNC2 neurons by leptin. a**, Representative images of a patched GFP-labeled BNC2 neuron. **b**, Spontaneous APs of BNC2 neurons in the ARC from ad-libitum-fed adult male BNC2-Cre mice (injected with DIO-GFP). Leptin was added to the bath solution at the indicated time window. **c**,**d**, Spontaneous AP frequency (**c**) and RMP (**d**) of BNC2 neurons before, during, and after leptin application (N = 12 cells from 4 mice, Friedman test or one-way ANOVA). **e**, Percentage of leptin-responsive BNC2 neurons under the overnight fasting and ad-libitum-fed conditions. Data are presented as mean ± SEM. *p < 0.05; **p < 0.01. Statistical details in Source Data.

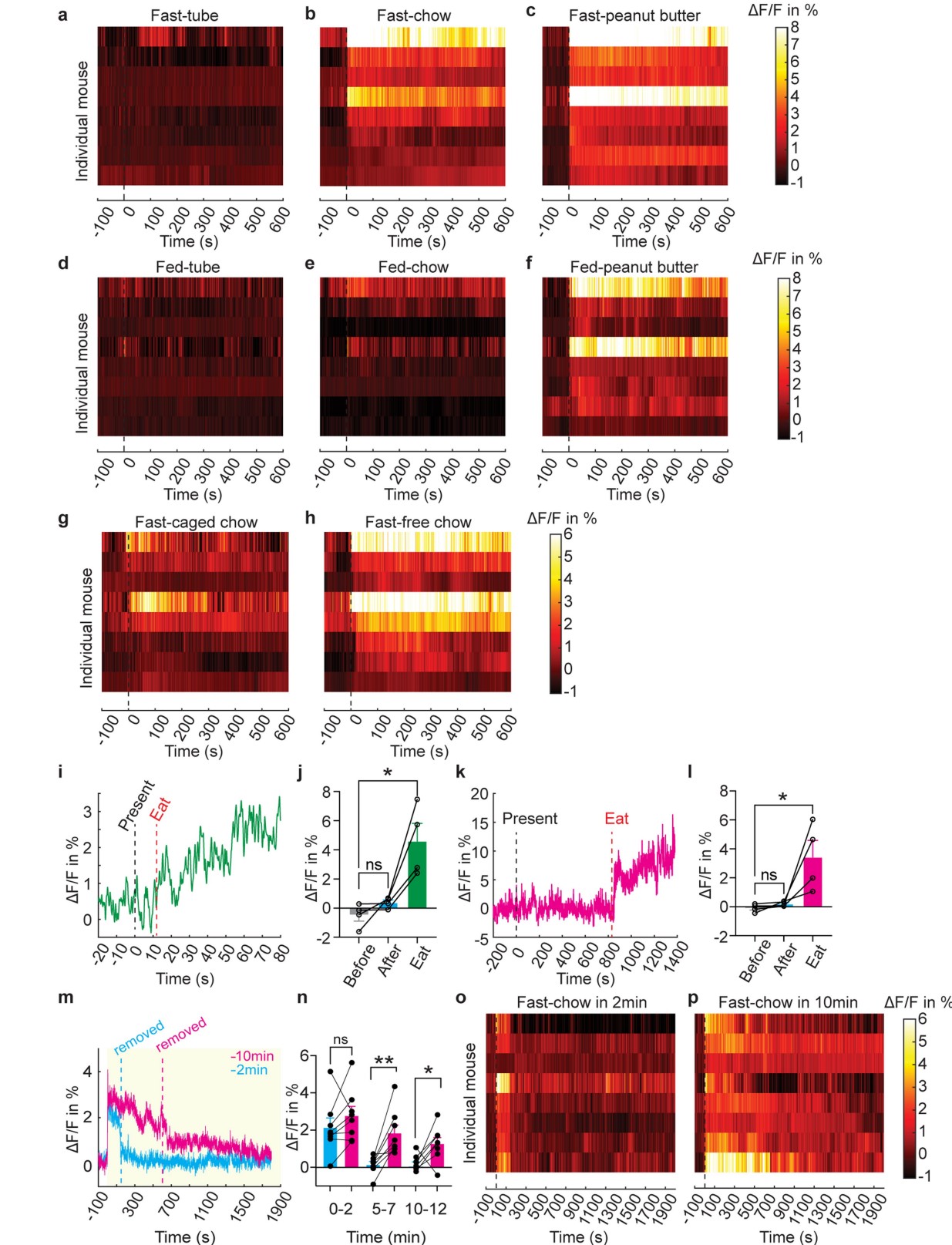

**Extended Data Fig. 4** | See next page for caption.

**Extended Data Fig. 4 | Rapid responses of BNC2 neurons to food cues.**
**a-c**, Heatmaps of normalized ΔF/F in individual overnight-fasted male mice aligned to the presentation of an inedible tube (**a**), chow (**b**), or peanut butter (**c**). **d-f**, Heatmaps of normalized ΔF/F in individual ad-libitum-fed male mice aligned to the presentation of an inedible tube (**d**), chow (**e**), or peanut butter (**f**). **g,h**, Heatmaps of normalized ΔF/F in individual overnight-fasted male mice aligned to the presentation of chow inside (**g**) or outside (**h**) the container. **i,k**, Individual calcium trace of an overnight-fasted naive male mouse presented with novel peanut butter (**i**) or sucrose tablets (**k**). **j,l**, Quantification of fluorescence changes before and after the presentation of peanut butter (**j**) or sucrose tablets (**l**), as well as during consumption (N = 4 mice, paired Student's *t*-test). **m-p**, Recordings from overnight-fasted male mice presented with chow for 2 or 10 min. **m**, Average calcium signal traces aligned to the time of presentation. **n**, Quantification of fluorescence changes in the indicated timeframes shown in **m** (N = 8 mice per group, paired Student's *t*-test). **o,p**, Heatmaps of normalized ΔF/F in individual overnight-fasted male mice aligned to the presentation of chow for 2 min (**o**) or 10 min (**p**). Data are presented as mean ± SEM. ns, not significant; *p < 0.05; **p < 0.01. Statistical details in Source Data.

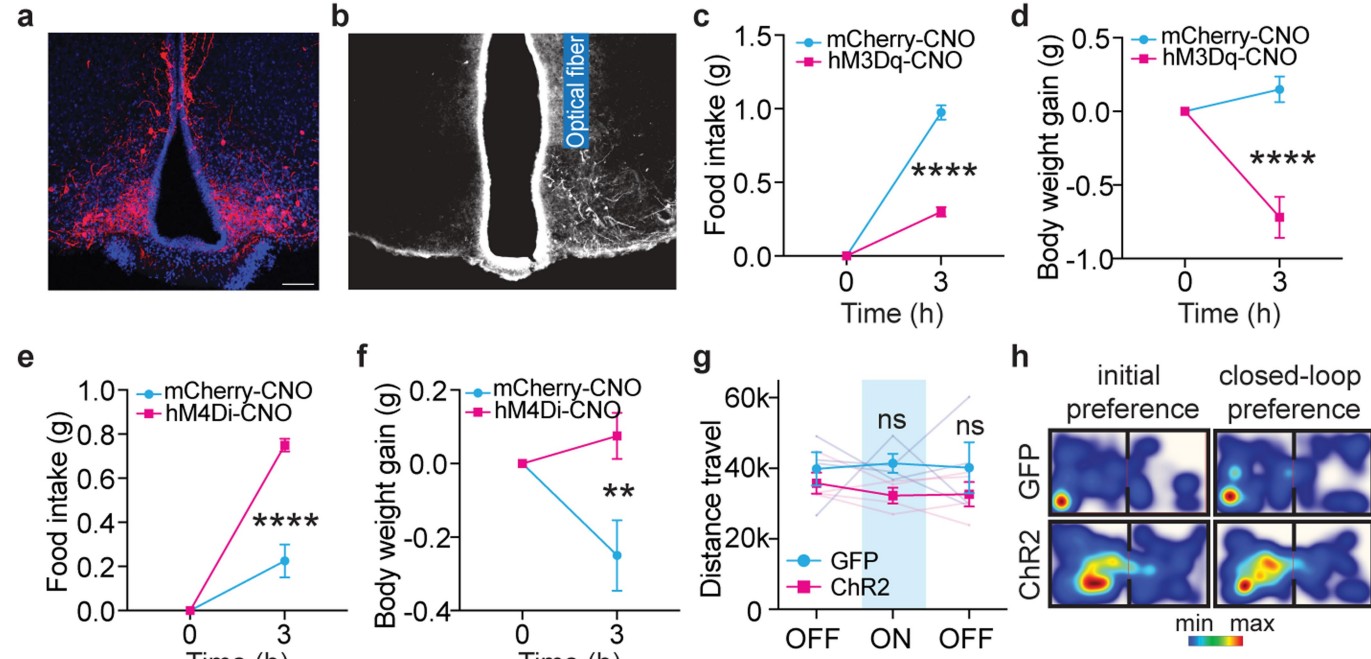

**Extended Data Fig. 5 | Rapid satiety driven by BNC2 neurons. a**, Representative image showing bilateral hM3Dq-mcherry expressing (red) in ARC BNC2 neurons with DAPI staining (blue) (N = 6 mice). Scale bar, 100 μm. **b**, Representative image showing unilateral ChR2-GFP expression in ARC BNC2 neurons and the optical fibre implant site (N = 12 mice). **c**, **d**, Food intake (**c**) and body weight change at 3 h (**d**) in female mice receiving PBS or CNO injection at the onset of the night dark phase (N = 4 mCherry-injected mice, N = 5 hM3Dq-injected mice, two-way ANOVA). **e**, **f**, Food intake (**e**) and body weight change at 3 h (**f**) in female mice receiving PBS or CNO injection during the daytime (N = 4 mCherry-injected mice, N = 4 hM4Di-injected mice, two-way ANOVA). **g**, Distance traveled before, during, and after laser stimulation in overnight-fasted male mice (N = 4 mice per group, two-way ANOVA). **h**, Heatmaps of time spent in each chamber during the initial phase and the closed-loop paired phase in sated adult male BNC2-Cre mice injected with GFP or ChR2. Data are presented as mean ± SEM. ns, not significant. **p < 0.01; ****p < 0.0001. Statistical details in Source Data.

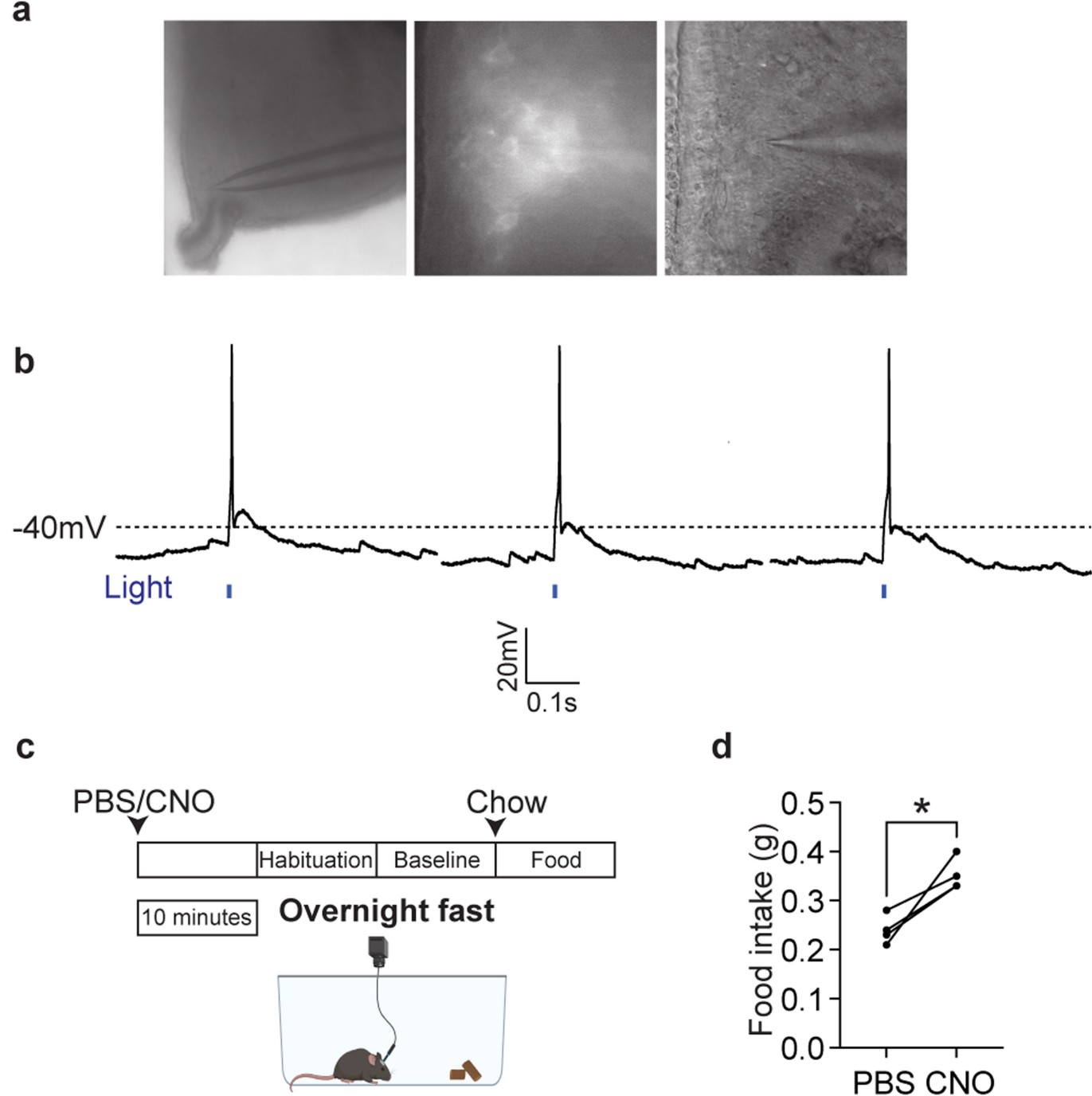

**Extended Data Fig. 6 | Light-induced activation of NPY neurons.**
**a**, Representative images of patched ChR2-GFP-labeled NPY neurons.
**b**, Detection of light-induced APs in ChR2-GFP-labeled NPY neurons.
**c**, Schematic of PBS or CNO administration timing during fibre photometry recordings of NPY neuron activity. **d**, Food consumption during the 10 min after food presentation in male mice injected with PBS or CNO (N = 4 mice, paired Student's *t*-test). *p < 0.05. Statistical details in Source Data. Illustration in **c** was created using BioRender (https://biorender.com).

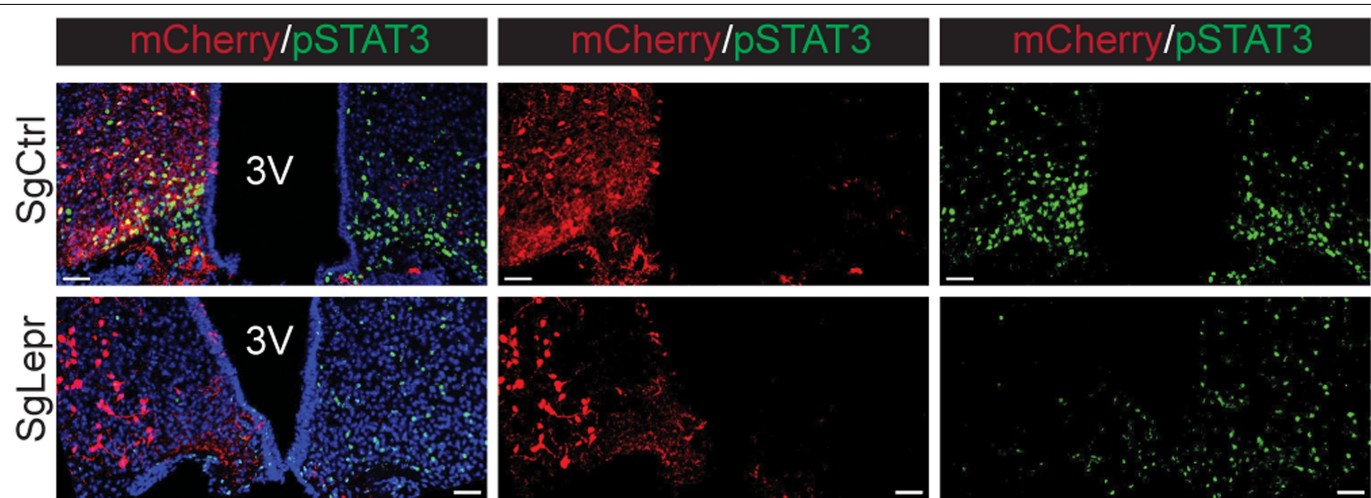

**Extended Data Fig. 7 | High efficiency of LepR deletion mediated by sgLepr.** Brain slices from overnight-fasted adult male BNC2-Cre::LSL-Cas9-GFP mice, injected unilaterally in the ARC with either sgCtrl or sgLepr, were analyzed 3 h post-leptin injection. The slices underwent mCherry and pSTAT3 immunostaining to assess the deletion efficiency (N = 3 mice per group). Scale bar, 50 μm.

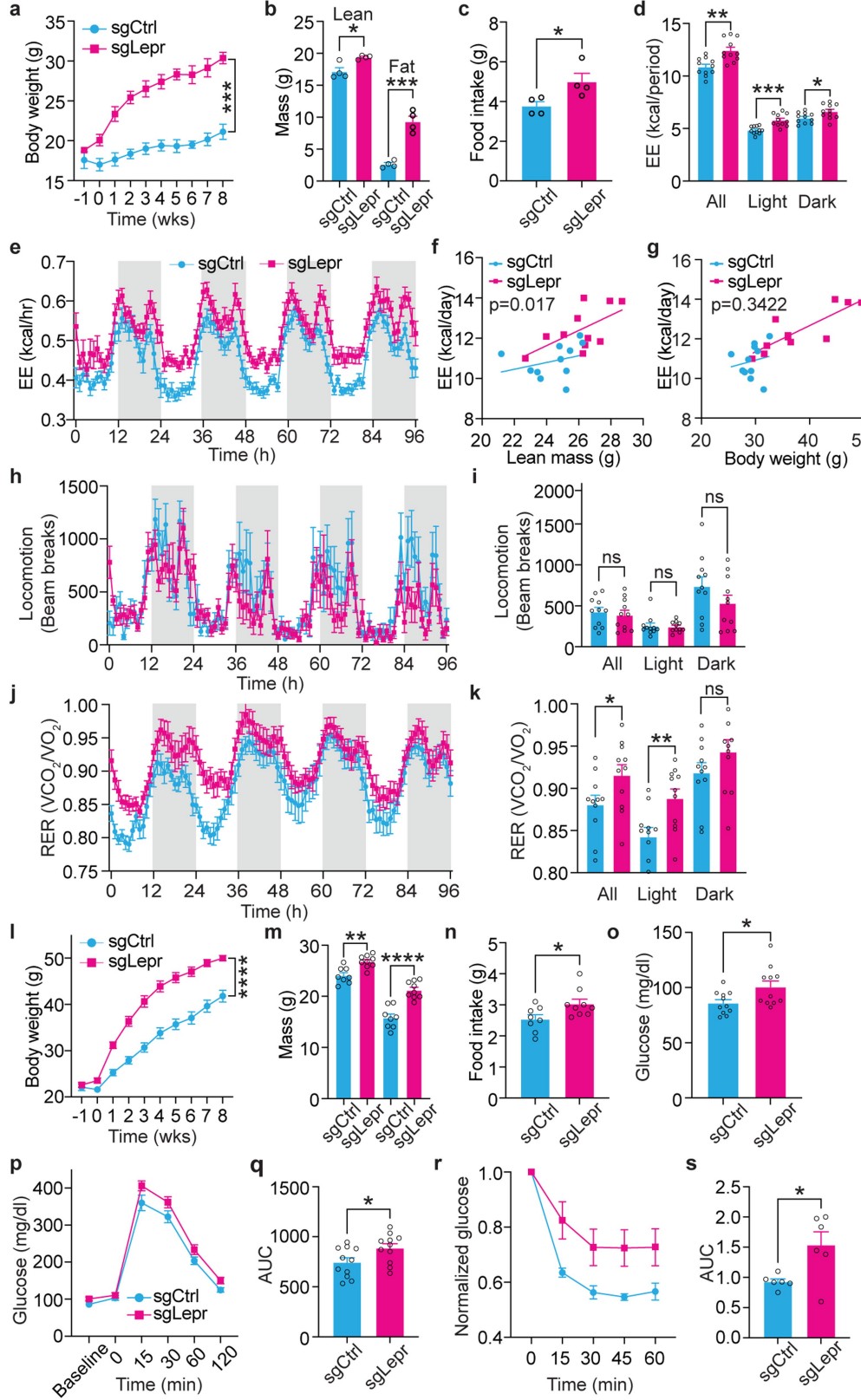

**Extended Data Fig. 8** | See next page for caption.

**Extended Data Fig. 8 | LepR knockout in BNC2 neurons causes obesity.**
**a**, Body weight of female mice on chow diet post sgCtrl or sgLepr injection
(N = 4 mice per group, two-way ANOVA). **b**, Body composition of female mice 8
weeks post-injection (N = 4 mice per group, unpaired Student's $t$-test). **c**, Daily
chow intake 8 weeks post-injection (N = 4 mice per group, unpaired Student's
$t$-test). **d**, **e**, EE om male mice from Fig. 5a measured by indirect calorimetry at
10–12 weeks post-injection (N = 11 mice per group, unpaired Student's $t$-test).
**f**,**g**, Regression analysis of EE vs. lean mass (**f**) and body weight (**g**) from **e** (N = 11
mice per group, ANCOVA). **h**,**i**, Locomotion for male mice from Fig. 5a (N = 11
mice per group, unpaired Student's $t$-test). **j**,**k**, RER in male mice from Fig. 5a
(N = 11 mice per group, unpaired Student's $t$-test). **l**, Body weight of male mice
on an HFD post sgCtrl or sgLepr injection (N = 8 sgCtrl-injected mice, N = 9
sgLepr-injected mice, two-way ANOVA). **m**, Body composition from I8 weeks
post-injection (N = 8 sgCtrl-injected mice, N = 9 sgLepr-injected mice, unpaired
Student's $t$-test). **n**, Daily HFD intake from I8 weeks post-injection (N = 8 sgCtrl-
injected mice, N = 9 sgLepr-injected mice, Mann-Whitney test). **o**, Fasting
glucose levels in male mice from Fig. 5a (N = 11 mice per group, unpaired Student's
$t$-test). **p**,**q**, GTT of male mice from Fig. 5a at 10–12 weeks post-injection (N = 11
mice per group, unpaired Student's $t$-test). **r**,**s**, ITT of male mice from Fig. 5a at
10–12 weeks post-injection (N = 6 mice per group, unpaired Student's $t$-test).
Data are presented as mean ± SEM. ns, not significant; *p < 0.05; **p < 0.01;
***p < 0.001, ****p < 0.0001. Statistical details in Source Data.

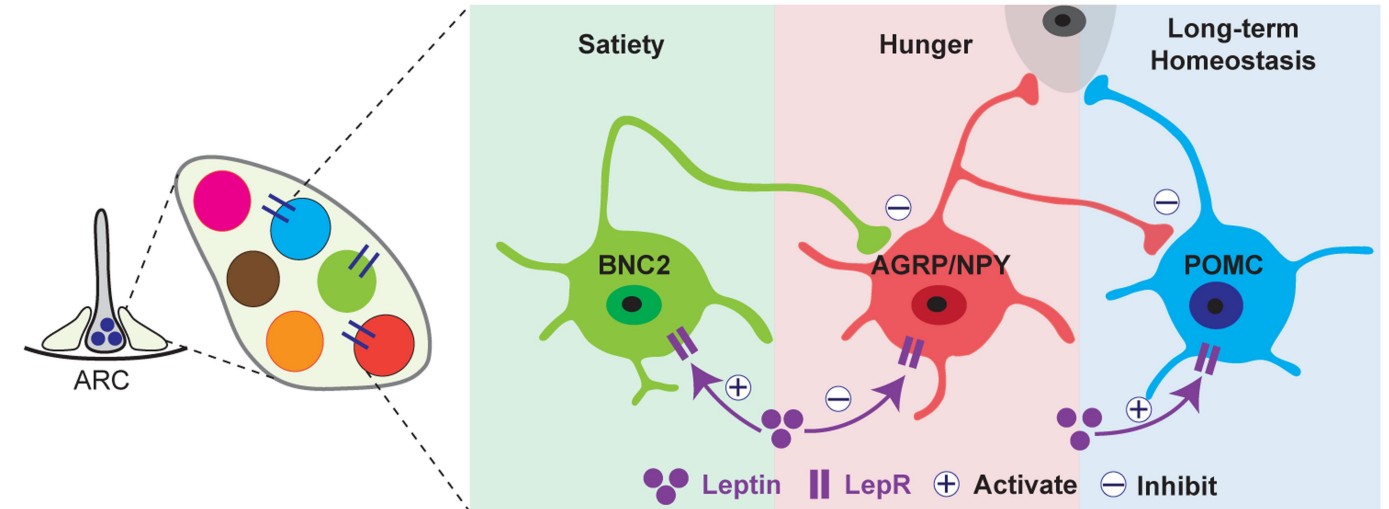

**Extended Data Fig. 9 | Model of leptin regulation of feeding and energy balance in ARC microcircuits.** Leptin-activated BNC2 neurons acutely promote satiety by suppressing AGRP/NPY neurons, which rapidly stimulates hunger. POMC neurons restrict weight gain by modulating energy balance over the long term with minimal short-term effects on food intake.

# Reporting Summary

## Statistics

For all statistical analyses, confirm that the following items are present in the figure legend, table legend, main text, or Methods section.

| n/a | Confirmed | |
|---|---|---|
| ☐ | ☒ | The exact sample size (*n*) for each experimental group/condition, given as a discrete number and unit of measurement |
| ☒ | ☐ | A statement on whether measurements were taken from distinct samples or whether the same sample was measured repeatedly |
| ☐ | ☒ | The statistical test(s) used AND whether they are one- or two-sided <br> *Only common tests should be described solely by name; describe more complex techniques in the Methods section.* |
| ☒ | ☐ | A description of all covariates tested |
| ☐ | ☒ | A description of any assumptions or corrections, such as tests of normality and adjustment for multiple comparisons |
| ☐ | ☒ | A full description of the statistical parameters including central tendency (e.g. means) or other basic estimates (e.g. regression coefficient) AND variation (e.g. standard deviation) or associated estimates of uncertainty (e.g. confidence intervals) |
| ☐ | ☒ | For null hypothesis testing, the test statistic (e.g. *F*, *t*, *r*) with confidence intervals, effect sizes, degrees of freedom and *P* value noted <br> *Give P values as exact values whenever suitable.* |
| ☒ | ☐ | For Bayesian analysis, information on the choice of priors and Markov chain Monte Carlo settings |
| ☒ | ☐ | For hierarchical and complex designs, identification of the appropriate level for tests and full reporting of outcomes |
| ☒ | ☐ | Estimates of effect sizes (e.g. Cohen's *d*, Pearson's *r*), indicating how they were calculated |

*Our web collection on statistics for biologists contains articles on many of the points above.*

## Software and code

Policy information about availability of computer code

| | |
|---|---|
| Data collection | Real-time place preference assay data were collected and analyzed using Ethovision XT13. Fiber photometry data were collected through the Fiber Photometry system from Tucker-Davis Technologies (RZ5P, Synapse) |
| Data analysis | The snRNA-seq data were analyzed using Cell Ranger version 5.0, Seurat v4 (v4.0.3), and R studio (2022.07.2). Matalab 2022b was used for the analysis of fiber photometry data. The analytic code is available at this GitHub repository: https://github.com/yuqiyuqitan/BNC2. |

For manuscripts utilizing custom algorithms or software that are central to the research but not yet described in published literature, software must be made available to editors and reviewers. We strongly encourage code deposition in a community repository (e.g. GitHub). See the Nature Portfolio guidelines for submitting code & software for further information.

## Data

Policy information about availability of data

All manuscripts must include a data availability statement. This statement should provide the following information, where applicable:
- Accession codes, unique identifiers, or web links for publicly available datasets
- A description of any restrictions on data availability
- For clinical datasets or third party data, please ensure that the statement adheres to our policy

All data generated or analyzed during this study are included in the manuscript and supporting files. The raw sequencing data have been deposited in the Gene Expression Omnibus and are available under the accession number GSE249564.

April 2023

# Research involving human participants, their data, or biological material

Policy information about studies with human participants or human data. See also policy information about sex, gender (identity/presentation), and sexual orientation and race, ethnicity and racism.

| | |
|---|---|
| Reporting on sex and gender | NA |
| Reporting on race, ethnicity, or other socially relevant groupings | NA |
| Population characteristics | NA |
| Recruitment | NA |
| Ethics oversight | NA |

Note that full information on the approval of the study protocol must also be provided in the manuscript.

# Field-specific reporting

Please select the one below that is the best fit for your research. If you are not sure, read the appropriate sections before making your selection.

☒ Life sciences    ☐ Behavioural & social sciences    ☐ Ecological, evolutionary & environmental sciences

For a reference copy of the document with all sections, see nature.com/documents/nr-reporting-summary-flat.pdf

# Life sciences study design

All studies must disclose on these points even when the disclosure is negative.

| | |
|---|---|
| Sample size | The sample size were determined based on similar studies in the filed (Xu et al., Nature. 2018. PMID: 29670283; Biglari et al., Nature Neuroscience 2021. PMID: 34002087; Chen et al., Cell.2015. PMID: 25703096; Aponte et al., Nature Neuroscience 2011. PMID: 21209617.). |
| Data exclusions | No data were excluded. |
| Replication | Experiments were performed independently across multiple animal replicates, as detailed in the paper. |
| Randomization | Mice were randomly assigned to control and experimental groups and littermates were used when possible. |
| Blinding | Behavioral data were collected and analyzed blind to whether animals were in control or experimental group. |

# Reporting for specific materials, systems and methods

We require information from authors about some types of materials, experimental systems and methods used in many studies. Here, indicate whether each material, system or method listed is relevant to your study. If you are not sure if a list item applies to your research, read the appropriate section before selecting a response.

## Materials & experimental systems

| n/a | Involved in the study |
|---|---|
| ☐ | ☒ Antibodies |
| ☒ | ☐ Eukaryotic cell lines |
| ☒ | ☐ Palaeontology and archaeology |
| ☐ | ☒ Animals and other organisms |
| ☒ | ☐ Clinical data |
| ☒ | ☐ Dual use research of concern |
| ☒ | ☐ Plants |

## Methods

| n/a | Involved in the study |
|---|---|
| ☒ | ☐ ChIP-seq |
| ☒ | ☐ Flow cytometry |
| ☒ | ☐ MRI-based neuroimaging |

# Antibodies

| | |
|---|---|
| Antibodies used | cFos antibody (1:1000, Synaptic systems, #226308), pSTAT3 antibody (1: 1000, Cell Signaling Technology, #9145s), GFP (1:1000, abcam, ab13970), RFP (1:1000, Rockland, #600-401-379), anti-NeuN Alexa Fluor 647-conjugated antibody (Abcam ab190565) (0.5 μl per million nuclei), and TotalSeq anti-Nuclear Pore Complex Proteins Hashtag antibody (BioLegend 682205) (0.5 mg per million |

nuclei).

| Validation | cFOS antibody (PMID: 31257028)<br>pSTAT3 antibody (PMID: 32699414)<br>GFP antibody (PMID:36413988)<br>RFP antibody (PMID: 35705049, 35659861)<br>NeuN antibody (PMID: 36071158, 12556490)<br>Nuclear pore complex (PMID: 12556490, 11448990) |
|---|---|

## Animals and other research organisms

Policy information about studies involving animals; ARRIVE guidelines recommended for reporting animal research, and Sex and Gender in Research

| Laboratory animals | Adult male and female (8-20 weeks) mice from strains: C57BL/6J (wild type; number 000664, The Jackson Laboratory), NPY-IRES2-FlpO-D (The Jackson Laboratory, stock no. 030211), Rosa26-LSL-Cas9 (The Jackson Laboratory, stock no. 024857), BNC2-P2A-iCre (made in-house). |
|---|---|
| Wild animals | This study did not involve wild animals |
| Reporting on sex | Some experiments were performed using both male and female mice and some were performed using male mice only. The sex information has been stated in the manuscript. |
| Field-collected samples | This study did not involve samples collected in the filed. |
| Ethics oversight | All animal experiments were approved by the Institutional Animal Care and Use Committee at Rockefeller University and were carried out in accordance with the National Institutes of Health guidelines. |

Note that full information on the approval of the study protocol must also be provided in the manuscript.

## Plants

| Seed stocks | NA |
|---|---|
| Novel plant genotypes | NA |
| Authentication | NA |

