## [Peer Review File · Nature]

Manuscript Title: Leptin Activated Hypothalamic BNC2 Neurons Acutely Suppress Food Intake

Reviewer Comments & Author Rebuttals

Reviewer Reports on the Initial Version:

Referees' comments:

Referee #1 (Remarks to the Author):

This manuscript identifies a small subpopulation of leptin-sensing neurons in the arcuate nucleus of the hypothalamus (ARC) that acutely suppresses appetite by directly inhibiting AgRP neurons. This population, marked by the expression of Bnc2, represents the missing piece of the puzzle in the field of homeostatic regulation of food intake. The prevailing model posits that the balance between opposing signals from leptin-sensing AgRP and POMC neurons in the ARC is a key determinant of feeding behavior. The validity of this simple model has been challenged by two types of observations: 1) Chemogenetic and optogenetic stimulation of POMC neurons to suppress feeding behavior acts on a much slower timescale than stimulation of AgRP neurons acts to induce feeding; and 2) Based on fiber photometry, leptin modulates the activity of AgRP and POMC neurons over hours and thus cannot explain leptin's acute anorectic effects.

While several groups have identified subpopulations of non-AgRP/POMC neurons in the ARC that express Lepr, the authors are the first to identify a unique molecular marker (Bnc2) that could be used to genetically target this population. Using a combination of immunohistochemical, electrophysiological and fiber photometry approaches, the authors demonstrate that BNC2 neurons are activated by leptin and the presence of food. The degree of activation is enhanced in the fasted state and in the context of a palatable diet. Importantly, chemogenetic and optogenetic modulation of BNC2 neurons altered feeding on a timescale that is similar to AgRP neurons. ChR2-assisted circuit mapping and fiber photometry experiments support the assertion that BNC2 neurons provide direct inhibitory signals to AgRP neurons that attenuate acute (10 min) food intake. Finally, CRISPR-mediated knockdown of Lepr in BNC2 neurons caused obesity due to hyperphagia, without deficits in energy expenditure. Together, these studies provide compelling evidence that BNC2 acutely suppresses food intake by inhibiting the activity of AgRP neurons and thus represents a major advance in the field. While the findings from complementary approaches support this hypothesis, the manuscript would be strengthened by better defining the role of LepR signals in BNC2 neurons in modulating acute responses to food. This is important, because a manuscript posted on BioRxiv claims that the anorectic effects of this population are mediated via Glp1R signaling.

Notable Issues:

1. Have any studies been performed in females? At the very least, it would be important to know whether BNC2 neurons are also present in females and whether knockdown of Lepr produces similar effects.
2. While most of the manuscript focused on the role of BNC2 neurons in acute responses to food, the authors only described chronic phenotypes in conditional Lepr KO mice. How do these mice respond

to leptin? A palatable diet?

3. As the authors discussed, it is not clear whether impaired glucose homeostasis caused by *Lepr* KD in BNC2 neurons is direct or secondary to obesity. Evaluation of the impact of chemogenetic inhibition of BNC2 neurons on glucose tolerance tests would be informative in this regard, as stimulation of AgRP neurons impairs glucose tolerance.

Minor Issues:

1. Line 61- insert the phrase “ablation of these neurons in adulthood has been reported to cause anorexia”.
2. Is labeling achieved by crossing the BNC2-Cre line to a tdTOM reporter specific to BNC2 neurons in the ARC or is there broader developmental expression? If expression is specific, what is the impact of deleting *Lepr* from birth by crossing to the floxed strain?
3. In the fiber photometry studies (Figure 3), was the degree of BNC2 neuronal activation negatively correlated with food intake?
4. In Figure 3 experiments using fiber photometry to monitor responses of BNC2 neurons to peanut butter, were the animals previously exposed to this food (to know it is palatable) or were they naïve? In the latter case, would they respond to non-nutritive pleasant aromas?
5. Figure 5a. It would be nice to have high magnification image of BNC2 projections relative to AgRP neurons. It is hard to see anything, because the red AgRP reporter is so strong.

Referee #2 (Remarks to the Author):

The identification of the *lepR*-expressing, BNC2 neurons in the arcuate hypothalamus and their role in regulation of food intake and body weight represents an important contribution to the field. The data are compelling and clearly described. I only have editorial suggestions:

Line 99, “thyrotropin releasing “ should be two words

Line 152, Most authors refer to GCaMP6 (as shown on line 566)

Lines 157-163, Figures and Legends, It would be better to use the Greek letter rather than spelling out Delta.

Line 223, 241, 252 Authors should use consistent nomenclature. BNC2-Cre without italics was the established abbreviation and should be used. Likewise, for the NPY construct. If italics is used, then the approved gene names should be used, *Bcn2* and *Npy*.

Line 265, “We assessed this possibility” would be better

Line 274 “Eight weeks after...” would be better

Line 308, I thought new data was being added with Ext Fig. 9; however, it looks like a nice summary instead. The sentence should be written.

Line 320, “Further studies, currently underway, knocking out” Would be better

Line 340, *Lep* ^{ob/ob} (italics) as superscript would be better

Line 395, “may directly influence” would be better

Line 404, “for this effect will be” would be better

Line 425, “(see below)” would be better than “in-house”

Line 534, 473-nm laser, add hyphen

Lines 536, 561 *Bcn2* should be BNC2 for consistency; *Npy* should be NPY

Line 560-562, leave spaces between wavelength and nm

Fig. 1, and ext data 1, It would be better to have all gene names in italics

Fig. 5, AAV should be added to the viral names

Ext Fig 2, legend Bnc2 mRNA should be in italics

Ext data 9, define the meaning of + and -

Author Rebuttals to Initial Comments:

Note: Reviewers' comments are in *blue italics* and our responses are in black.

Referees' comments:

Referee #1 (Remarks to the Author):

This manuscript identifies a small subpopulation of leptin-sensing neurons in the arcuate nucleus of the hypothalamus (ARC) that acutely suppresses appetite by directly inhibiting AgRP neurons. This population, marked by the expression of Bnc2, represents the missing piece of the puzzle in the field of homeostatic regulation of food intake. The prevailing model posits that the balance between opposing signals from leptin-sensing AgRP and POMC neurons in the ARC is a key determinant of feeding behavior. The validity of this simple model has been challenged by two types of observations: 1) Chemogenetic and optogenetic stimulation of POMC neurons to suppress feeding behavior acts on a much slower timescale than stimulation of AgRP neurons acts to induce feeding; and 2) Based on fiber photometry, leptin modulates the activity of AgRP and POMC neurons over hours and thus cannot explain leptin's acute anorectic effects.

We appreciate the reviewer's comment that our identification of BNC2 neurons "represents the missing piece of the puzzle in the field of homeostatic regulation of food intake."

While several groups have identified subpopulations of non-AgRP/POMC neurons in the ARC that express Lepr, the authors are the first to identify a unique molecular marker (Bnc2) that could be used to genetically target this population. Using a combination of immunohistochemical, electrophysiological and fiber photometry approaches, the authors demonstrate that BNC2 neurons are activated by leptin and the presence of food. The degree of activation is enhanced in the fasted state and in the context of a palatable diet. Importantly, chemogenetic and optogenetic modulation of BNC2 neurons altered feeding on a timescale that is similar to AgRP neurons. ChR2-assisted circuit mapping and fiber photometry experiments support the assertion that BNC2 neurons provide direct inhibitory signals to AgRP neurons that attenuate acute (10 min) food intake. Finally, CRISPR-mediated knockdown of Lepr in BNC2 neurons caused obesity due to hyperphagia, without deficits in energy expenditure. Together, these studies provide compelling evidence that BNC2 acutely suppresses food intake by inhibiting the activity of AgRP neurons and thus represents a major advance in the field. While the findings from complementary approaches support this hypothesis, the manuscript would be strengthened by better defining the role of LepR signals in BNC2 neurons in modulating acute responses to food. This is important, because a manuscript posted on BioRxiv claims that the anorectic effects of this population are mediated via Glp1R signaling.

The reviewer's description of our studies is accurate, and we appreciate the statement that the studies "provide compelling evidence that BNC2 (neurons) acutely suppresses food intake by inhibiting the activity of AgRP neurons" and is a "major advance". The reviewer also raises an interesting question with respect to data on GLP1 signaling in the report by Webster et al which we have addressed in a set of new studies described below¹.

We investigated the role of LepR and Glp1R signaling in BNC2 neurons by ablating BNC2 neurons in the ARC and evaluating the response to leptin and semaglutide, a Glp1R agonist. The neurons were ablated by injecting an AAV-mCherry-flex-dtA bilaterally into the ARC of adult BNC2-Cre mice. Mice injected with AAV-DIO-mCherry served as controls. Five weeks after the injection, we administered leptin (5 mg/kg), semaglutide (10 nmol/kg), or vehicle (PBS) to these mice at the beginning of the dark cycle and measured their food intake over 3 hours. Both leptin and semaglutide significantly suppressed food intake in the control mice, with semaglutide having a greater effect (see **Response Fig. 1** below). In mice with BNC2 neuron ablation, the effect of leptin to reduce food intake was significantly diminished compared to mice that were injected with the mCherry construct. In contrast, the response to semaglutide was similar in the dtA-injected vs. mCherry-injected mice (**Response Fig. 1**). These results suggest that BNC2 neurons play a role in mediating the acute effects of leptin on feeding, but not those of the Glp1R agonist. We have added these new data in the revised manuscript (see **Figure Fig. 7e**). Webster et al find that the TRH neurons they studied respond to GLP-1 but not leptin. Thus these studies indicate that the BNC2 population we identified is functionally distinct from the TRH-expressing neurons reported by Webster et al.

Response Fig. 1: Ablating BNC2 neurons blunted the acute response to leptin but not semaglutide. We ablated BNC2 neurons by injecting an AAV with a floxed dtA into the ARC of BNC2-Cre mice. Control mice received injections of a floxed mCherry construct. Food intake of mice was measured 3 hours after mice received injections of PBS, leptin (5 mg/kg), or semaglutide (Sema, 10 nmol/kg) at the beginning of the dark phase.

Notable Issues:

1. Have any studies been performed in females? At the very least, it would be important to know whether BNC2 neurons are also present in females and whether knockdown of *LepR* produces similar effects.

As suggested, we have repeated the studies in female mice. Consistent with our observations in male mice, chemogenetic activation of BNC2 neurons at the beginning of the dark cycle in female mice led to a significant reduction in food intake and body weight compared to control mice (**Response Fig. 2a,b**). Similar to males, silencing BNC2 neurons during the light cycle significantly increased food consumption and body weight in females (**Response Fig. 2c,d**). Finally, a knockout of *LepR* in BNC2 neurons in adult females caused hyperphagia and increased body weight relative to control mice (**Response Fig. 2e,g**). Similar to what we observed in males, fat mass was greatly increased in the sgLepr females with a small increase in lean mass relative to the sgCtrl group (**Response Fig. 2f**). Collectively, these results show that BNC2 neurons function similarly in males and females. We have added these new data in the revised manuscript (see **Extended Data Fig. 5c-f, Extended Data Fig. 8a-c**).

Response Fig. 2: Function of BNC2 neurons in females. a,b, Food intake and body weight change over 3 hours of each group of female mice receiving PBS or CNO injection upon entering the night dark phase. c, d, Food intake and body weight change over 3 hours of each group of female mice receiving PBS or CNO injection in the daytime. e, Body weight of female mice fed on chow following injection of sgCtrl or sgLepr. f Body

composition of two groups of female mice 8 weeks after virus injection. **g**, Daily chow intake of two groups 8 weeks after virus injection.

2. While most of the manuscript focused on the role of BNC2 neurons in acute responses to food, the authors only described chronic phenotypes in conditional LepR KO mice. How do these mice respond to leptin? A palatable diet?

We have assessed the role of BNC2 neurons in the response to leptin and a palatable diet as follows. We first knocked out LepR in BNC2 ARC neurons by injecting AAVs carrying sgLepr bilaterally into the ARC of adult BNC2-Cre::LSL-Cas9-GFP mice. Mice injected with sgCtrl served as controls. We then assessed the response to PBS and leptin 2 weeks after virus injection, prior to the time when the LepR knockout mice became obese. PBS or leptin (5 mg/kg) was administered to sgCtrl-injected or sgLepr-injected mice immediately before the onset of the dark phase and food intake was measured over 3 hours. We found that the mice with a LepR knockout in BNC2 neurons consumed significantly more chow compared to sgCtrl-injected mice after PBS injections (**Response Fig. 3a**). Similar to the above (see **Response Fig. 1**), the effect of leptin to reduce food intake was significantly decreased on the mice with a LepR knockout in BNC2 neurons (**Response Fig. 3a**). Finally, we assessed the response to a palatable diet by feeding the BNC2-LepR knockout mice a high-fat diet (Research Diets, 60 kcal% fat). We found that mice with a LepR knockout in BNC2 neurons gained significantly more weight compared to the sgCtrl-injected mice (**Response Fig. 3b**). Fat mass within the sgLepr group was also significantly increased relative to the sgCtrl group with a small increase in lean mass (**Response Fig. 3c**). The knockout mice also consumed significantly more of the high-fat diet compared to controls (**Response Fig. 3d**). We have included these data in the revised manuscript (see **Fig. 7d**, **Extended Data Fig. 8 I-n**).

Response Fig. 3: Knockout LepR in BNC2 neurons blunted leptin effect and increased HFD intake. a, Food intake at 3 hours of each group of male mice receiving PBS or leptin injection upon entering the night dark phase. **b**, Body weight of male mice fed on a high-fat diet (HFD) following injection of sgCtrl or sgLepr. **c**, Body composition of two groups at 8 weeks after virus injection. **d**, Daily HFD intake of two groups at 8 weeks after virus injection.

3. As the authors discussed, it is not clear whether impaired glucose homeostasis caused by LepR KD in BNC2 neurons is direct or secondary to obesity. Evaluation of the impact of chemogenetic inhibition of BNC2 neurons on glucose tolerance tests would be informative in this regard, as stimulation of AgRP neurons impairs glucose tolerance.

As suggested by the reviewer, we have investigated the effect of BNC2 neuron activity on glucose metabolism using chemogenetics. To standardize conditions and avoid confounding effects of differences in daily food intake, mice were fasted overnight and food was not provided throughout the experiments. AAVs carrying a DIO-Gq-DREADDs, or DIO-mCherry, were injected bilaterally into the ARC of BNC2-Cre mice. Activating BNC2 neurons by administering CNO for 1 hour significantly decreased blood glucose levels in mice that received Gq-DREADDs injections compared to control mCherry-injected mice (**Response Fig. 4a**). We also observed an improvement in glucose tolerance and enhanced insulin sensitivity in Gq-injected mice compared to mCherry-injected controls (**Response Fig. 4b-e**). This is the opposite effect that activating AgRP neurons has². In contrast, chemogenetic silencing of BNC2 neurons increased blood glucose levels and worsened glucose tolerance and insulin sensitivity (**Response Fig. 4f-j**). These data show that BNC2 neurons also regulate peripheral glucose

homeostasis independent of their acute effect on feeding. We have added these results in the revised manuscript (see **Fig. 7f-o**).

Response Fig. 4: BNC2 neurons regulate glucose metabolism. AAVs with floxed activating or inhibitory DREADDs were injected in the ARC of BNC2-Cre mice. CNO was injected into these animals and controls and the following studies were performed after an overnight fast. **a**, Glucose levels of two groups at 0, 1 h upon CNO injection following 16 h overnight fasting. **b,c**, Glucose tolerance test (GTT) of two groups of mice at 1h after CNO injection. **d,e**, Insulin tolerance test (ITT) of two groups of mice at 1h after CNO injection. **f**, Glucose levels of two groups at 0, 1 h upon CNO injection following 16 h overnight fasting. **g,h**, GTT of two groups of mice at 1h after CNO injection. **i,j**, ITT of two groups of mice at 1h after CNO injection.

Minor Issues:

1. Line 61- insert the phrase “ablation of these neurons in adulthood has been reported to cause anorexia”.

We have inserted the phrase as suggested by the reviewer.

2. Is labeling achieved by crossing the BNC2-Cre line to a tdTOM reporter specific to BNC2 neurons in the ARC or is there broader developmental expression? If expression is specific, what is the impact of deleting *LepR* from birth by crossing to the floxed strain?

Bnc2 is a transcription factor that is expressed during brain development and a global knockout leads to embryonic lethality^{3,4}. While it is expressed in several brain regions in adults, its expression in ARC is limited to the LepR population we describe. We confirmed this by injecting AAV-DIO-mCherry into the ARC of adult BNC2-Cre mice and confirmed that the mCherry reporter was specifically expressed only in BNC2 neurons in the ARC (**Extended Data Fig. 2d**). Because *Bnc2* is also expressed outside the ARC in the DMH, cortex, hippocampus, and thalamus of adult mice, mating BNC2-Cre to floxed *LepR* mice would potentially lead to knockouts in multiple brain regions^{5,6}. For this reason, we studied the function of LepR signaling in ARC BNC2 neurons by deleting it specifically in ARC BNC2 neurons by injecting AAV viruses carrying sgLepr into the ARC of the adult mice. Crossing the BNC2-Cre mouse line with the floxed *LepR* mouse line would not distinguish among possible effects in other regions. We discussed the rationale of this study in detail in our revised manuscript (see Paragraph 3 of the Discussion).

3. In the fiber photometry studies (Figure 3), was the degree of BNC2 neuronal activation negatively correlated with food intake?

In the initial fiber photometry studies, we observed a rapid activation of ARC BNC2 neurons following the presentation of food, with a plateau prior to the time that the mice began consuming the food (**Response Fig.5a**).

To investigate the impact of food intake on BNC2 neuronal activation, we also compared BNC2 neural activity before, during, and after food consumption. We analyzed the responses using peanut butter, as it is more straightforward to determine whether the mice were actively eating or not as compared to the chow pellet which mice sometimes chew without consuming it. We found that the activity of BNC2 neurons further increased during the consummatory phase and was significantly higher than the signal prior to this. The activity of these neurons returned to baseline when consumption ceased (**Response Fig.5b-e**). These results indicate that BNC2 neurons are activated by food cues and further activated by food consumption. We have added these data in the revised manuscript (see **Fig. 3g-j**). Thus in response to the reviewer's comment, the activity of the BNC2 neurons in ARC is positively correlated with food intake.

Response Fig. 5: Food consumption increased BNC2 neuron activity. **a**, Individual trace of BNC2 neuron activity in one mouse when presented with peanut butter after an overnight fast. **b,d**, Average traces of calcium signals in fasted mice represented with peanut butter aligned to the time of eating. **c,e**, Quantification of fluorescence changes within the 3-second timeframe before, after, and during food consumption.

4. In Figure 3 experiments using fiber photometry to monitor responses of BNC2 neurons to peanut butter, were the animals previously exposed to this food (to know it is palatable) or were they naïve? In the latter case, would they respond to non-nutritive pleasant aromas?

As the reviewer surmised, mice were pre-exposed to peanut butter so that they would recognize it as a palatable food. To evaluate this further, we monitored their responses to peanut butter in naïve mice that had never encountered it before. In contrast to the mice that had previously been exposed to peanut butter and consumed it immediately, there was a far longer latency of naïve mice to start eating (data not shown). Consistent with this, during the exploratory period when the peanut butter was present to naïve mice but had not yet been consumed, we did not observe any change in BNC2 neuron activity (**Response Fig. 6a,b**). BNC2 neuron activity then increased significantly once the mice consumed the food (**Response Fig. 6a,b**). As an alternative, we also presented high-sucrose tablets to the naïve mice and monitored the changes in BNC2 neuron activity. Here again, there was no response when the pellets were first provided but BNC2 neuron activity increased when the mice began eating (**Response Fig. 6c,d**). These data demonstrate that the response of BNC2 neurons to a novel food cue is experience-dependent. We have added these data in the revised manuscript (see **Extended Data Fig. 4g-j**).

Response Fig. 6: BNC2 neurons did not respond to novel food cues. **a,c**, Individual calcium trace of the overnight-fasted naïve mouse presented with a novel peanut butter (**a**) or sucrose tablets (**c**). **b,d**, Quantification of fluorescence changes before and after peanut butter (**b**) or sucrose tablet (**d**) presentation, as well as during consumption.

5. Figure 5a. It would be nice to have high magnification image of BNC2 projections relative to AgRP neurons. It is hard to see anything, because the red AgRP reporter is so strong.

Response: We have replaced the previous images with higher magnification ones in the revised manuscript (see Fig. 5a).

Referee #2 (Remarks to the Author):

*The identification of the *lepR*-expressing, BNC2 neurons in the arcuate hypothalamus and their role in regulation of food intake and body weight represents an important contribution to the field. The data are compelling and clearly described. I only have editorial suggestions:*

We appreciate the reviewer's statement that our results are "an important contribution to the field" and that "the data are compelling and clearly described." We address their specific comments below.

Line 99, "thyrotropin releasing " should be two words

We have corrected that in the revised manuscript.

Line 152, Most authors refer to GCaMP6 (as shown on line 566)

We have used GCaMP6s throughout the revised manuscript as suggested.

Lines 157-163, Figures and Legends, It would be better to use the Greek letter rather than spelling out Delta.

We have used the Greek letter in the revised manuscript.

*Line 223, 241, 252 Authors should use consistent nomenclature. BNC2-Cre without italics was the established abbreviation and should be used. Likewise, for the NPY construct. If italics is used, then the approved gene names should be used, *Bcn2* and *Npy*.*

We apologize for the inconsistency and have removed the italics, ensuring consistent nomenclature in the revised manuscript.

Line 265, "We assessed this possibility" would be better

We have changed the sentence as suggested.

Line 274 "Eight weeks after..." would be better

We have edited this line as suggested.

Line 308, I thought new data was being added with Ext Fig. 9; however, it looks like a nice summary instead. The sentence should be written.

We have edited the sentence accordingly as suggested and rephrased it as Summary Fig.1.

Line 320, "Further studies, currently underway, knocking out" Would be better

We have changed the sentence as suggested.

Line 340, Lep ob/ob (italics) as superscript would be better

We have changed this as suggested.

Line 395, “may directly influence” would be better

As suggested by Reviewer 1, we have directly tested the effect of ARC BNC2 neurons on glucose metabolism. The results of these new studies are shown below. To standardize conditions and avoid confounding effects of differences in daily food intake, mice were fasted overnight and food was not provided throughout the experiments. AAVs carrying a DIO-Gq-DREADDs, or DIO-mCherry, were injected bilaterally into the Arc of Bnc2-cre mice. Activating BNC2 neurons by administering CNO for 1 hour significantly decreased blood glucose levels in mice that received Gq-DREADDs injections compared to control mCherry-injected mice (**Response Fig.4a**). We also observed an improvement in glucose tolerance and enhanced insulin sensitivity in Gq-injected mice compared to mCherry-injected controls. This is the opposite effect that activating AGRP neurons has² (**Response Fig.4b-e**). In contrast, chemogenetic silencing of BNC2 neurons increased blood glucose levels and worsened glucose tolerance and insulin sensitivity (**Response Fig.4f-j**). These data show that BNC2 neurons also regulate peripheral glucose homeostasis independent of their acute effect on feeding. We have added these results in the revised manuscript (see **Fig. 7f-o**).

Response Fig. 4: BNC2 neurons regulate glucose metabolism. AAVs with floxed activating or inhibitory DREADDs were injected in the Arc of Bnc2 neurons. CNO was injected into these animals and controls and the following studies were performed after an overnight fast. **a**, Glucose levels of two groups at 0, 1 h upon CNO injection following 16 h overnight fasting. **b,c**, Glucose tolerance test (GTT) of two groups of mice at 1h after CNO injection. **d,e**, Insulin tolerance test (ITT) of two groups of mice at 1h after CNO injection. **f**, Glucose levels of two groups at 0, 1 h upon CNO injection following 16 h overnight fasting. **g,h**, GTT of two groups of mice at 1h after CNO injection. **i,j**, ITT of two groups of mice at 1h after CNO injection.

Line 404, “for this effect will be” would be better

We have changed the sentence as suggested.

Line 425, “(see below)” would be better than “in-house”

We have changed the sentence as suggested.

Line 534, 473-nm laser, add hyphen

We have added a hyphen as suggested.

Lines 536, 561 Bcn2 should be BCN2 for consistency; Npy should be NPY

We have corrected this throughout the revised manuscript.

Line 560-562, leave spaces between wavelength and nm

We have corrected that as suggested.

Fig. 1, and ext data 1, It would be better to have all gene names in italics

We have changed this as suggested.

Fig. 5, AAV should be added to the viral names

We have added AAV to the viral names as suggested.

Ext Fig 2, legend Bnc2 mRNA should be in italics

We have edited this as suggested.

Ext data 9, define the meaning of + and –

'+' indicates 'activate' while '-' shows 'inhibit'. We have clarified that in the revised manuscript as suggested.

References

- 1 Webster, A. N. *et al.* Molecular Connectomics Reveals a Glucagon-Like Peptide 1 Sensitive Neural Circuit for Satiety. *bioRxiv*, 2023.2010.2031.564990 (2023). <https://doi.org:10.1101/2023.10.31.564990>
- 2 Steculorum, S. M. *et al.* AgRP Neurons Control Systemic Insulin Sensitivity via Myostatin Expression in Brown Adipose Tissue. *Cell* **165**, 125-138 (2016). <https://doi.org:10.1016/j.cell.2016.02.044>
- 3 Vanhoutteghem, A. *et al.* Basonuclin 2 has a function in the multiplication of embryonic craniofacial mesenchymal cells and is orthologous to disco proteins. *Proc Natl Acad Sci U S A* **106**, 14432-14437 (2009). <https://doi.org:10.1073/pnas.0905840106>
- 4 Vanhoutteghem, A. *et al.* The importance of basonuclin 2 in adult mice and its relation to basonuclin 1. *Mech Dev* **140**, 53-73 (2016). <https://doi.org:10.1016/j.mod.2016.02.002>
- 5 Yao, Z. *et al.* A high-resolution transcriptomic and spatial atlas of cell types in the whole mouse brain. *Nature* **624**, 317-332 (2023). <https://doi.org:10.1038/s41586-023-06812-z>
- 6 Zhang, M. *et al.* Molecularly defined and spatially resolved cell atlas of the whole mouse brain. *Nature* **624**, 343-354 (2023). <https://doi.org:10.1038/s41586-023-06808-9>

Reviewer Reports on the First Revision:

Referees' comments:

Referee #1 (Remarks to the Author):

The authors performed additional experiments that addressed all of my concerns. Key points that are clarified/resolved include: 1) BNC2 neurons are functionally distinct from TRH+/GLP1R+ neurons; 2) effects of manipulating BNC2 neuronal activity or Lepr expression are not sex-specific; 3) BNC2 neurons modulate glycemic control, independent of body weight; and 4) loss of Lepr from BNC2 neurons blunts the anorexic effect of leptin and increases susceptibility to HFD. While the BNC2 neurons do not account for all of leptin's effects on feeding, this study provides compelling evidence that they play an important role in regulating energy balance.